# Global floating kelp forests have limited protection despite intensifying marine heatwave threats

Nur Arafeh-Dalmau[1,2,3,4,5] ✉, Juan Carlos Villaseñor-Derbez [1,6,7], David S. Schoeman[8,9], Alejandra Mora-Soto[5,10], Tom W. Bell [11], Claire L. Butler [12], Maycira Costa[10], Loyiso V. Dunga[5,13,14,15], Henry F. Houskeeper[11], Cristian Lagger[16,17], Carolina Pantano[18], Daniela Laínez del Pozo [19], Kerry J. Sink [14,15], Jennifer Sletten[20], Timothe Vincent[20], Fiorenza Micheli [1,21] & Kyle C. Cavanaugh[2]

Kelp forests are one of the earth's most productive ecosystems and are at great risk from climate change, yet little is known regarding their current conservation status and global future threats. Here, by combining a global remote sensing dataset of floating kelp forests with climate data and projections, we find that exposure to projected marine heatwaves will increase ~6 to ~16 times in the long term (2081–2100) compared to contemporary (2001–2020) exposure. While exposure will intensify across all regions, some southern hemisphere areas which have lower exposure to contemporary and projected marine heatwaves may provide climate refugia for floating kelp forests. Under these escalating threats, less than 3% of global floating kelp forests are currently within highly restrictive marine protected areas (MPAs), the most effective MPAs for protecting biodiversity. Our findings emphasize the urgent need to increase the global protection of floating kelp forests and set bolder climate adaptation goals.

Marine protected areas (MPAs) are a cornerstone of marine conservation[1]. Promoted by international agreements, such as the Convention on Biological Diversity (CBD) Aichi Target 11[2], the area of marine ecosystems under some form of protection has increased since the turn of the century[3]. Because climate change is a major long-term threat to biodiversity[4–6], the newly agreed Global Biodiversity Framework at COP15[7] calls for effectively protecting 30% of the oceans by 2030. A central component of the post-2020 targets is increasing the representation of different habitats under effective protection while adapting to climate change. Although many studies report the protection of critical habitat-forming species, such as corals, seagrass, and mangroves[3], other essential marine habitats, such as kelp forests, remain largely neglected[8] (but see refs. 9,10). Comprehensive maps on kelp forest distribution, threats associated with climate change,

extreme events, and protection status are urgently needed to guide ongoing local and global protection efforts.

Kelp forests dominate >30% of the world's rocky reefs and are among the most productive ecosystems on earth—comparable to terrestrial forests[11–14]. However, marine heatwaves (MHWs) and anthropogenic activities threaten kelp forests[15–18] and their capacity to provide ecosystem services worth billions of dollars[19–22]. The IPCC Sixth Assessment Report identified kelp forests as the second most at-risk marine ecosystem from MHWs[6], after coral reefs, which is concerning given that MHWs are projected to become more frequent and severe in the next decades[23]. In addition, kelp forests are facing other stressors (e.g., pollution, overgrazing) that can reduce their ability to recover from heatwaves. For example, northern California has lost >90% of its kelp forests due to the combined effects of severe marine

heatwaves and overgrazing by sea urchins[24,25]. Climate adaptation strategies−including MPAs−are urgently needed to halt and reverse this loss[16,26,27]. While MPAs cannot directly counter the impacts of climate change that can surpass a species' physiological tolerance[28], MPAs can mitigate non-climatic stressors like overfishing and habitat destruction, which can enhance ecosystem resilience[29,30], supporting ecological functioning and providing societal benefits[31–33].

Well-managed and highly restrictive MPAs−no-take marine reserves where all fishing activities are prohibited−are the most effective type of MPA for rebuilding fished populations[31], supporting the stability of kelp forest ecosystems[34] and, in some documented cases, providing resilience to MHW impacts[35–38]. For example, in regions where urchin predators are protected from fishing and where trophic cascades are a driver of food-web dynamics, MPAs can facilitate the recovery of higher-trophic-level organisms, which helps control kelp grazer populations and prevent overgrazing of kelp[30,39,40]. This mechanism has been found to support resistance to and recovery from MHWs in a network of 39 MPAs in southern California[30]. In addition, MPAs can provide climate resilience for kelp forests ecosystems through other mechanisms[35–37,41–43]. For example, a recent global analysis found that fish communities were more stable to MHW in highly restrictive MPAs than unprotected sites[37], and abalone populations in two MPAs in Mexico were more resilient to a hypoxia event and MHWs through increased body size and reproductive output[36,42].

Monitoring subtidal kelp populations over large spatial and temporal scales can be challenging. However, the largest species (i.e., *Macrocystis pyrifera, Nereocystis leutkeana, Ecklonia maxima*) can be mapped by remote sensing because they create extensive canopies that float on the water surface. Recent advances in satellite imaging of surface-canopy-forming kelp species provide an opportunity to map the distribution of kelp forest habitats, quantify the threats posed by MHWs, and assess their protection status[44]. Floating kelp forests are a globally distributed foundation species that co-exist with other sub-canopy kelp species that structure one of Earth's most biodiverse ecosystems[12]. These forests can cover thousands of hectares (e.g., 28,500 hectares in the Southern California Bight ecoregion[10]) and sustain hundreds to thousands of species, some of which are economically and culturally significant. For example, in the Channel Islands in Southern California, studies found 716 species associated with giant kelp forest ecosystems[45] and in Patagonia, Chile and Argentina, similar studies found between 150 and 250 species[46–48]. In addition, in the northeast Pacific Ocean, 17 species of sub-canopy kelp coexist with floating kelp forests (www.algaebase.org/) from Alaska to Baja California, Mexico. Since remote sensing is the only available method to detect kelp forest ecosystems comprehensively (i.e., using standarized methods at large temporal and spatial scales), maps of floating kelps are good indicators of the broader ecosystem and associated biodiversity. These data can also inform other climate-adaptation strategies such as identifying and protecting climate refugia[49,50]−areas less impacted by or more resilient to climate change −for kelp forests. Effectively protecting climate refugia for kelp forest ecosystems is a priority for conservation[51] because, in these areas, biodiversity can persist[49] and may enhance the resilience of other kelp forests, depending on local and regional connectivity and life history-traits, by maintaining a source of recovery for impacted kelp habitats[26].

Here, we compile a comprehensive global map of floating kelp forest habitats (henceforth "kelp forests") and leverage these datasets to project the global exposure of kelp forests to MHWs and asses their protection status within MPAs. To develop the global kelp forest map, we assemble existing regional and national remote-sensing datasets from Landsat observations (1984–present), supplemented with Sentinel-2 satellite imagery (2015–2019[52]) (Supplementary Table 1; see methods). To project threats to kelp forests from climate change, we estimate future cumulative annual MHW intensities from an ensemble of sea surface temperature (SST) from 11 Earth System models, using three climate scenarios generated under the IPCC Shared Socio-Economic Pathways (SPPs)[53] (see methods). We then quantify the global protection status and the representation of kelp forests at both country and biogeographic levels (i.e., realm, ecoregions[54]) within MPAs categorized as highly, moderately, or less protected based on restrictions to extractive activities obtained from ProtectedSeas[55] (see methods). Our findings reveal increasing threats to all floating kelp forests from future MHWs, although some southern hemisphere forests may act as climate refuges. We also found that kelp forests remain largely unprotected within restrictive MPAs, the most effective type of MPA, which are poorly represented globally. These findings emphasize the urgent need to increase the global protection and effective representation of kelp forests and, given the scale of the threat posed by future MHWs, for bolder climate adaptation goals for kelp forests.

## Results
### Global distribution of kelp forests
We found floating kelp forest habitats in only 12 nations distributed across 6 biogeographic realms and 32 ecoregions, mostly in mid-latitudes in the Pacific, Atlantic, and Indian Oceans (Fig. 1a). Most of the kelp forests are located in five ecoregions, with 23.7% in Malvinas/Falklands, 20.9% in Channels and Fjords of Southern Chile, 12.8% in Southern California Bight, 10.3% in Kerguelen Islands, and 9.2% in Northern California; while 17 ecoregions combined account for only 1% of the distribution of kelp forests (Supplementary Fig. 1).

In the northern hemisphere, kelp forests can be found at their highest latitudes, overall, in the USA (~61.4 °N), extending southward to their warm-distribution limit in Mexico (~27 °N). In the southern hemisphere, kelp forests can be found at their lowest latitudes, overall, in Peru (~13.6 °S), extending southwards to their cool-distribution limit in Chile (~56 °S). Other warm-distribution limits of kelp forests in the southern hemisphere are located in Argentina, Namibia, South Africa, Australia, and New Zealand.

### Contemporary and Future exposure of kelp forests to marine heatwaves
Projected future MHWs for kelp forests increase for each realm, ecoregion, climate scenario, and time (Figs. 1b and 2 and Supplementary Figs. 2 and 3). In the near term (2021-2040), kelp forests are projected to be subject to > 2 times higher exposure to cumulative MHW intensities compared to contemporary exposure, with similar values across climate scenarios (Supplementary Table 2-5). Projections suggest that these magnitudes will continue to intensify, and under SSP5-8.5, kelp forests could be subject to >6 to >16 times higher cumulative MHW intensities in the mid (2041–2060) and long term (2081–2100), respectively, compared to contemporary exposure (Supplementary Table 6). These magnitudes are ~2 to ~3 times higher than corresponding projections under SSP1-2.6 and SSP2-4.5, respectively. In the long term, even under SSP1-2.6 and SSP2-4.5, magnitudes are ~5.6 and ~9.6 times higher than contemporary exposure, respectively. Note that these estimations where derived from the mean cumulative MHW intensities (n = 2156 pixels) for each climate scenario and time frame, and then divided by the corresponding mean contemporary values.

The Arctic and the Temperate North Pacific realms are projected to be the most exposed to future MHWs under all climate scenarios, while Temperate South America and Temperate South Africa will be the least exposed (Fig. 1b), matching the general spatial patterns in contemporary exposure. Overall, the pattern is very similar across SSP scenarios, with the northern hemisphere experiencing nearly twice the exposure to future MHWs than the southern hemisphere (Fig. 2b). However, some differences emerge. We found a difference in the latitudinal pattern of exposure between the northern and southern hemisphere. Specifically, projections suggest a latitudinal pattern of increasing exposure to future MHWs from lower to higher latitudes in

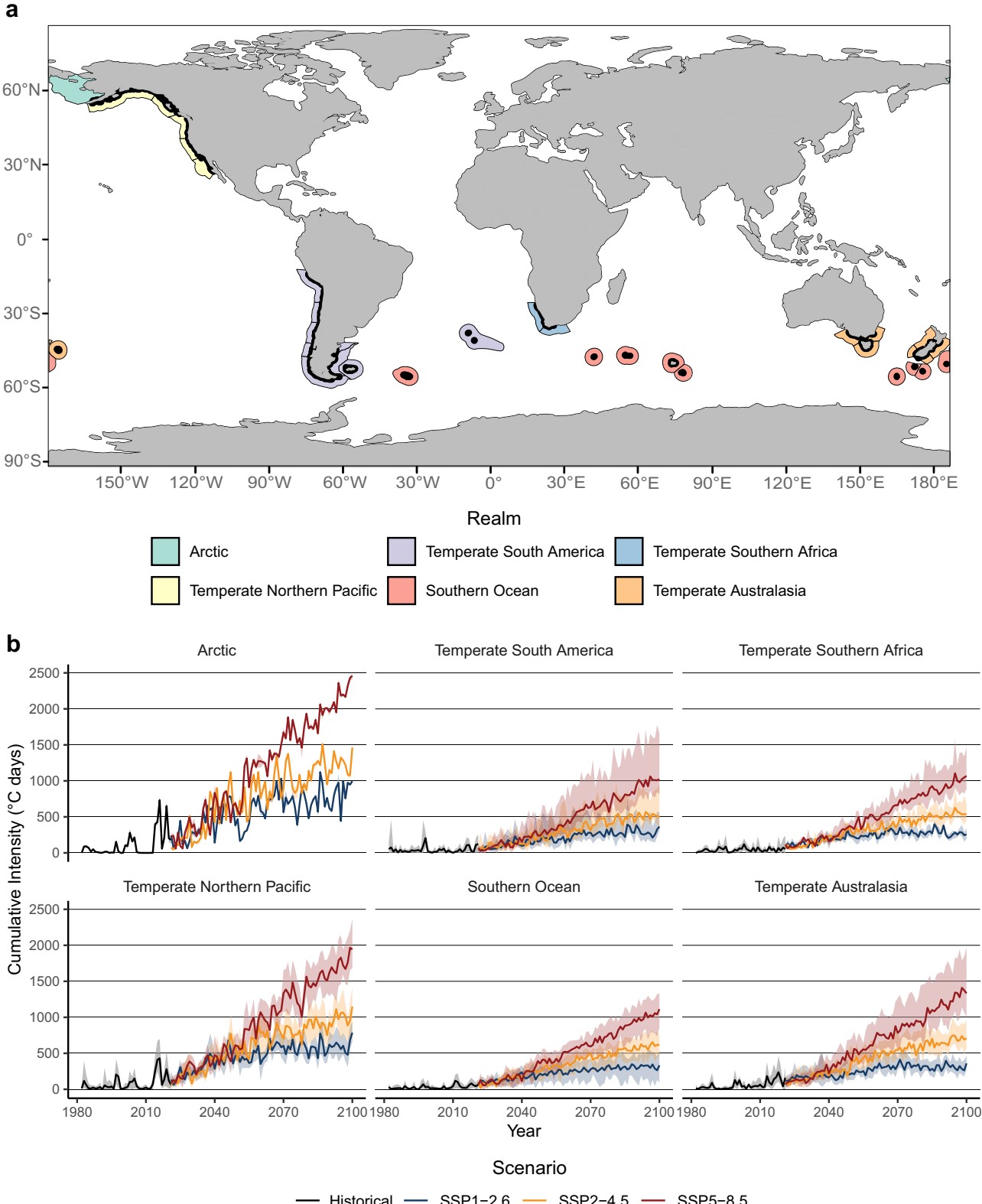

**Fig. 1 | Global distribution of floating kelp forests and exposure to contemporary and future marine heatwaves.** Panel (**a**) map of kelp distribution (black lines) across 32 biogeographic ecoregions (census[54]) (polygons; the color indicates the realm to which they belong), (**b**) realm-specific exposure of kelp forest to historical (1982–2020) and future cumulative annual marine heatwave intensities (2021–2100) across three climate scenarios under IPCC Shared Socio-Economic Pathways (SSP1-2.6, SSP2-4.5, SSP5-8.5). The solid line shows the mean across ensemble medians for all pixels, and the shaded area represents the 5th and 95th percentiles.

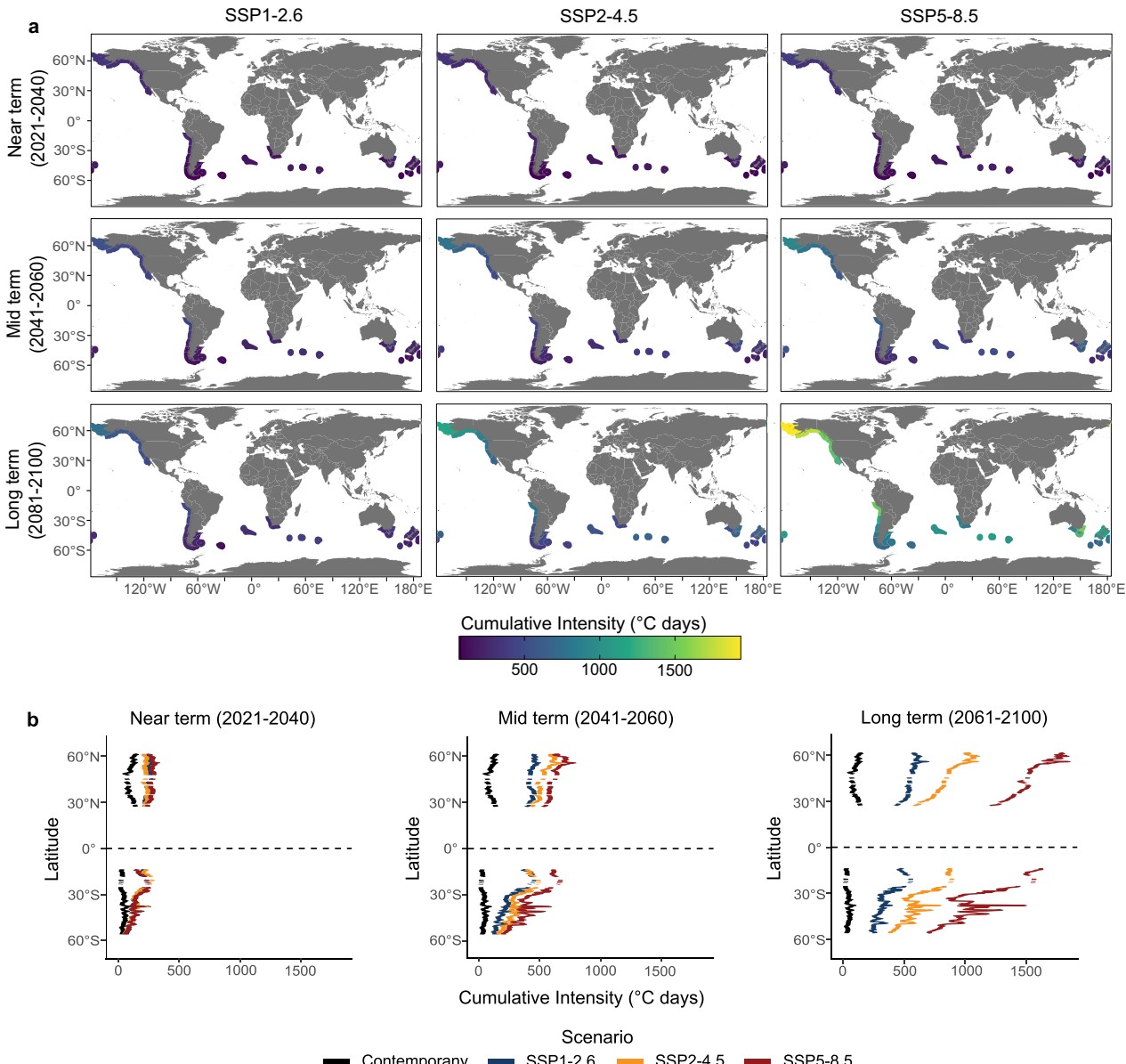

**Fig. 2 | Ecoregional exposure of floating kelp forests to contemporary and future marine heatwaves.** Panel (**a**) mean future cumulative annual marine heatwave intensity for all pixels in each of 32 ecoregions across three climate scenarios under IPCC Shared Socio-Economic Pathways (SSP1-2.6, SSP2-4.5, SSP5-8.5) and three-time frames (near, mid, and long term). **b** Latitudinal plots representing mean cumulative annual marine heatwave intensities by 1° of latitude under contemporary (2001–2020) and climate scenarios for each time.

the northern hemisphere, whereas in the southern hemisphere, this pattern is reversed (Fig. 2b). For example, in the mid and long term and under all future scenarios for the northern hemisphere, the Eastern Bering Sea and the Gulf of Alaska are projected to become the most exposed ecoregions, while the southern California Bight becomes the least exposed (Fig. 2a and Supplementary Fig. 3), albeit with elevated levels of MHW exposure relative to the present. In contrast, in the southern hemisphere lower latitude ecoregions such as Cape Howe and Humboldtian are projected to be the most exposed to future MHWs while remote islands in high latitudes and ecoregions such as the Channels and Fjords of Southern Chile will be the least exposed.

### Global protection status of kelp forests
Globally, more than 33.1% of floating kelp forest habitats are protected by MPAs, of which 13.7% are highly protected (the most effective type of MPA), 4.6% are moderately protected, and 14.8% are in less-

protected MPAs (Figs. 3a, b and 4a). However, most of the effective protection for kelp forests is in remote islands in the Southern Ocean realm (24,319.8 ha), and when excluding these areas, only 2.8% (5,870.9 ha) of the global kelp forests are highly protected from fishing activities (Fig. 3c). At the country level, France has placed all their floating kelp forests within highly protected MPAs (Fig. 4a, b) and is the only country that meets the current 30% effective representation target[7]. New Zealand, South Africa, Canada, Australia, and the USA have at least 10% of their kelp forests highly protected (Fig. 4a, b). However, this protection is in overseas territories in remote islands for all of France (23,007.1 ha, there is no floating kelp forests in mainland France) and much of New Zealand (145.1 ha), South Africa (285.1 ha), and Australia (78.7 ha). Australia has only 2.7% (23.0 ha), New Zealand 2.0% (14.1 ha), and South Africa (400.6 ha) 8.9% of their continental kelp forests highly protected. Mexico and the UK have provided

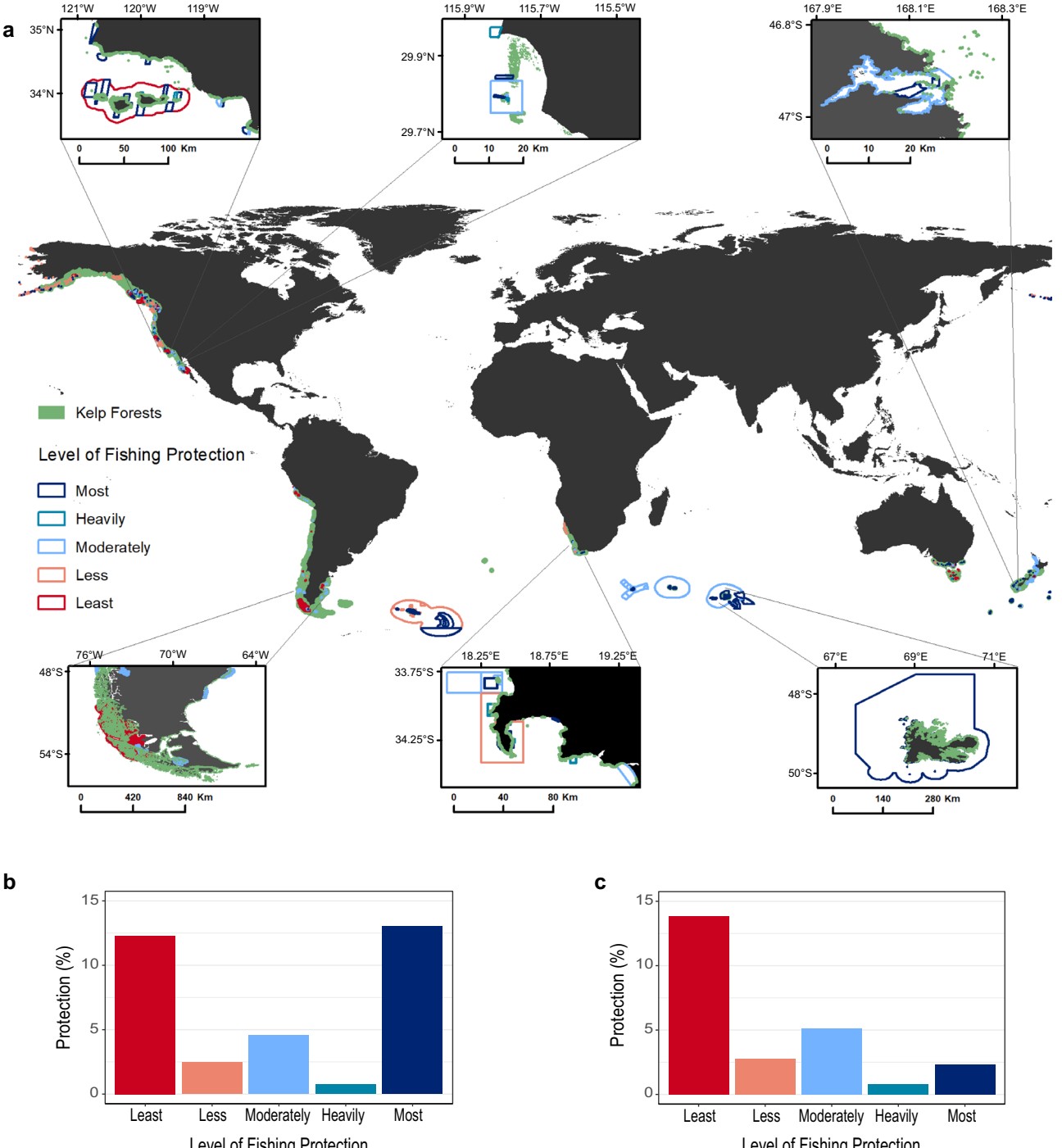

**Fig. 3 | Global distribution of floating kelp forests and marine protected areas by categories of protection. a** Global map of kelp forests and marine protected areas, we provide six fine-scale views. Starting from the top-left and moving clockwise: USA, Mexico, New Zealand, France (Kerguelen and Crozet Islands), South Africa, and Chile and Argentina. Global protection (%) of kelp by category of protection (**b**) including all realms and (**c**) excluding the Southern Ocean realm. Protection categories are based on the Level of Fishing Protection (LFP)[55] score assigned to each marine protected area. The scores are divided in three categories: Lightly protected (LFP score of "Least" and "Less"), moderately protected, and highly protected (LFP score of "Heavily" and "Most"). Global basemap boundaries were derived from the ESRI ArcGIS World Countries Generalized shapefile.

effective protection for less than 2% of their kelp forests, Chile less than 0.02%, and Peru, Argentina, and Namibia none.

Of the world's biogeographic realms, the Southern Ocean realm has 99.9% of its kelp forests within highly protected MPAs (which represents 11% of the global distribution, see Figs. 3c and 4), while all other realms have less than 10%. However, at least 10% of kelp forests are protected in some form of MPA in all realms, except for the Arctic,

where the area of surface-canopy forming kelp is minimal and no kelp forests are protected under any category (Fig. 4c). At the ecoregional level, only 9 ecoregions have met the old 10% effective representation targets[2] for kelp forests within highly protected MPAs, all in remote islands except for the Northern California ecoregion (Supplementary Fig. 4). Overall, 47.2% of ecoregions have less than 10% of their kelp forests protected, regardless of the MPA type. Only one nation, one

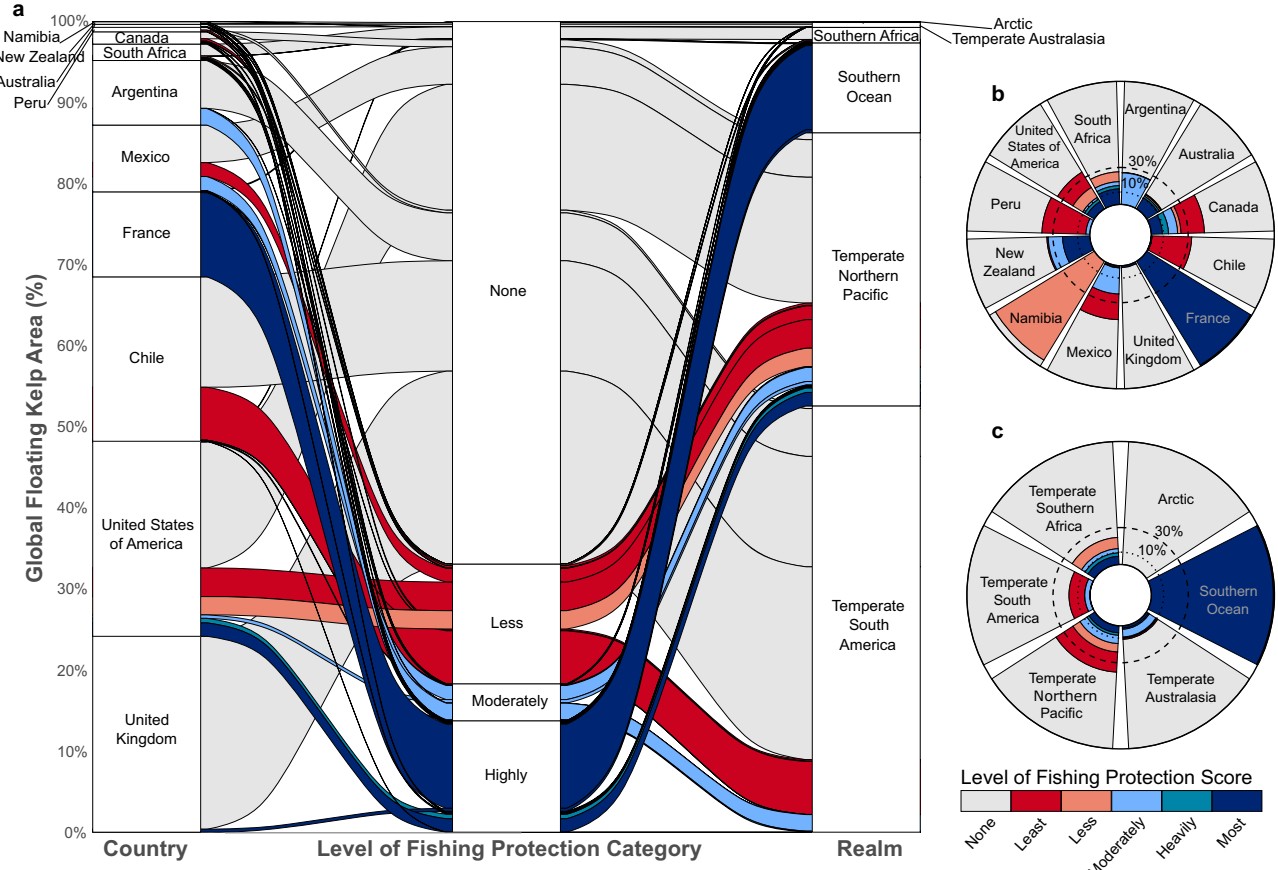

**Fig. 4 | Global status and distribution of floating kelp forest protection.**
**a** Alluvial diagram with the distribution and protection of kelp by country and realm (% of total area), and radial plots showing percentage protection of kelp at the level of (**b**) country and (**c**) biogeographic realm. Protection categories are based on the Level of Fishing Protection (LFP)[55] score assigned to each marine protected area. The dotted and dashed lines show the old 10%[2] and the current 30%[7] effective protection targets. Note that France and the United Kingdom have no floating kelp forests in their mainland and are all in their overseas territories in the Southern Ocean realm. We included the Malvinas/Falkland Islands as part of the United Kingdom territory, although we acknowledge that Argentina has ongoing legal claims for their sovereignty.

realm, and 25% of ecoregions (all remote islands) meet the new 30% target for effective representation[7] for kelp forests.

### Ecoregional future marine heatwave threats and protection status
Kelp forests within the ecoregions that are most threatened by projected MHWs and currently have low levels of effective protection (highly protected) include the Bering Sea (none protected), the Gulf of Alaska (0.6%), the North American Fjordlands (2.5%), the Puget Trough (0.09%), and the Oregon to Vancouver ecoregions (2.4%) (Fig. 5a, b and Supplementary Figs. 5 and 6). Northern California is the only ecoregion projected to be highly threatened by MHWs where at least 10% of kelp forests are inside highly protected MPAs. In contrast, eight ecoregions that have all their kelp forests inside highly protected MPAs will face low to intermediate threats from projected MHWs under the SSP2-4.5 scenario. These ecoregions are all located in remote islands of the Southern Ocean realm. When combining highly and moderately protected MPAs, the Patagonian Shelf and North Patagonian Gulfs ecoregions have at least 30% of their kelp forests protected and low exposure to MHWs (Fig. 5b).

### Discussion
We present a global map of the protection status of floating kelp forest habitats, which allowed us to identify escalating climate change threats and important conservation gaps for kelp forest ecosystems globally. Although one nation and a few ecoregions are meeting current

international protection targets[7] for kelp forests, many of these MPAs are in remote islands with low levels of exposure to contemporary and projected MHWs and few non-climatic threats[56]. When kelp forests in remote islands are excluded, less than 3% of kelp forests are inside highly restrictive MPAs—no-take marine reserves—the most effective type of MPA for conserving biodiversity[1,31] and for enhancing climate resilience[29,30,35–37,39,42]. Thus, current global protection does not adequately account for anthropogenic pressures on kelp forest ecosystems. It is concerning that the kelp forests most exposed to current and projected MHWs have minimal protection, which suggests that their resilience is likely being compromised. Therefore, to achieve international conservation commitments and climate adaptation goals, most countries and ecoregions require additional investments to increase the area of kelp forest habitats that are effectively protected. This presents a unique opportunity for designing and implementing climate-smart MPAs[26].

Our study reveals that marine heatwaves will increasingly threaten kelp forests under all projected SSP scenarios and time frames. If greenhouse emissions are not mitigated, kelp forests could be exposed to >16 times the magnitude of contemporary exposure under extreme scenarios by the end of the century. That represents an increase of 2–5 °C in average ocean temperatures, which in some regions may permanently surpass physiological tolerances of kelp forests, impact their distribution, restructure associated ecological communities and impact the livelihood of local human communities[4,15,17,19,20,57–60]. Note that our study assessed the exposure to

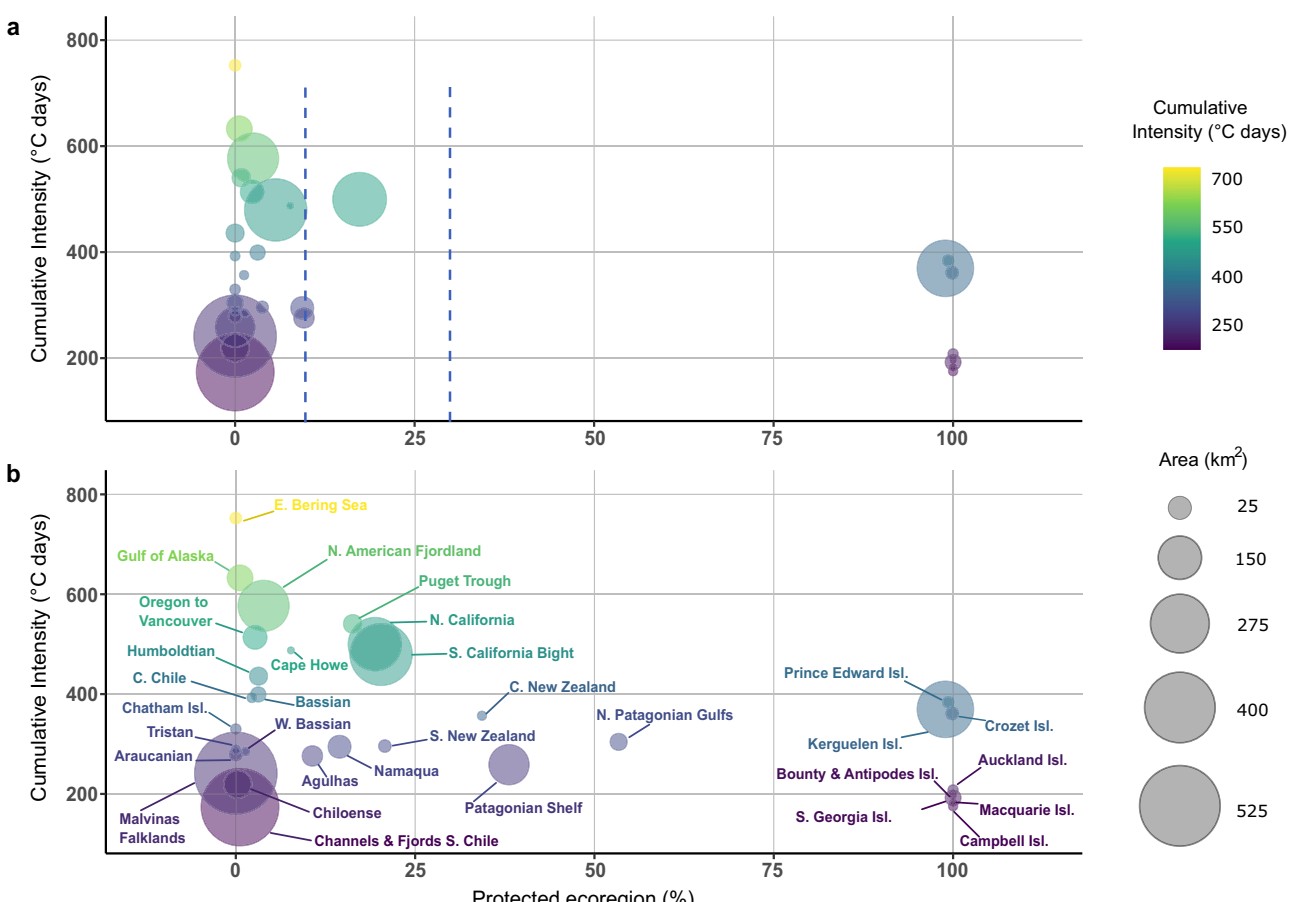

**Fig. 5 | Relationship between threat posed by future marine heatwaves and level of protection for floating kelp forests. a** Scatterplots of mean future cumulative annual marine heatwave intensities for the midterm (2041–2060) under an intermediate IPPC Shared Socio-Economic Patway (SSP2-4.5) climate scenario and the amount of kelp forests highly protected, (**b**) highly and moderately protected combined. The size of the bubble indicates the amount of kelp in each ecoregion. The dashed blue vertical lines represent the old 10%[2] and the current 30%[7] targets for effective protection.

projected MHWs, not the vulnerability of kelp forests. While kelp forests near their current warm distribution limit will likely be the most affected and subject to range contractions[15,58,61,62], populations living further from their thermal tolerance limit may be less threatened, and novel climates may favor kelp expansion[58,59]. However, cold-range populations subject to extreme MHWs could also be threatened because these populations may be less adapted to extreme temperatures[63]. Moreover, the persistence of kelp forest ecosystems is not driven solely by available substratum and suitable temperatures but also by biotic interactions[64], and not all components of these ecosystems will necessarily respond in the same way. Predicting whether MPAs can provide resilience to kelp forest ecosystems under such extreme and persistent changes is therefore challenging on multiple fronts. However, for less-extreme emission scenarios that track current mitigation policies[65,66], the magnitude of exposure to future MHWs will be two times lower than for extreme scenarios. Under these conditions, it is more likely that marine reserves can support the resilience of kelp forest ecosystems.

MPAs cannot directly mitigate the impacts of MHWs that surpass the physiological thresholds of kelp forests; however, they can minimize other non-climatic threats, such as overfishing and habitat destruction, thereby promoting the recovery of kelp forests following MHWs. For example, after the 2014–2016 MHWs in the northeast Pacific Ocean, urchins overgrazed kelp forests and caused many of them to collapse into less biodiverse ecosystems[16,25]. Studies in the Channel islands have shown that urchin barrens reduced sessile invertebrate diversity by 40% and almost completely lost canopy fish

assemblage[45]. However, highly protected MPAs have promoted the resilience of kelp forests following MHWs by facilitating recovery of overfished predators that control urchin populations[30,40]. Because the magnitude of future warming may cause the loss of kelp forests in some regions, MPAs will likely not be enough on their own to support the persistence of kelp forests. In these cases, supplementary climate-adaptation strategies will be necessary, particularly for areas of high exposure to future MHWs, such as regions in North America and especially areas near-warm distribution limits. These strategies include identifying and protecting climate refugia, restoring degraded kelp, identifying genetically resilient kelp stocks, and managing other anthropogenic impacts (e.g., land-based pollution) not mitigated by MPAs[16,67].

We identified areas that will likely act as climate refugia—projected to be less exposed to future MHWs—where kelp forests are likely to persist[9,10,26,68]. We found that although many ecoregions with potential climate refugia have all their kelp forests protected inside MPAs, the Southern Fjordlands of Chile and the Malvinas/Falklands ecoregions have no protection and account for >40% of the global distribution of kelp forests. These ecoregions emerge as priority areas for global conservation of kelp forests, and efforts are needed to secure their effective protection and representation[56] before other non-climatic threats intensify and erode their resilience.

It is important to note that our analysis maps floating kelp forests, thus our method will not detect other kelp forest ecosystems when they do not co-occur with floating kelps. There are > 120

laminarian kelp species (many are intertidal), of which three of the largest kelp species form extensive floating canopies that can be detected by remote sensing, including the globally distributed *M. pyrifera*. Our estimates likely represent overall kelp forest distribution and ecosystem protection in regions where floating kelps co-exist with other sub-canopy kelp species (e.g., the west coast of North America, South America, and remote southern hemisphere islands, among others). However, some other nations and regions not included here have extensive kelp forest ecosystems that do not co-occur with floating kelp forests (the east coast of North America, northern Europe, and parts of Australia, among others). Given the limitations in detecting subsurface canopy kelp forests, they are likely less-well represented here than those detected by remote sensing. This is a substantial gap for kelp conservation and an avenue for novel technologies and research[69] to address associated needs for these kelp ecosystems that do not overlap with floating kelps, which also support diverse and productive ecosystems[13,70] and human livelihoods[21].

We also note that our compiled map may underestimate floating kelp habitat for those regions where regional maps are not yet available (e.g., Canada, Chile, New Zealand) because the global map covers a shorter time period than the regional Landsat data and so may miss kelp habitat that was not present between 2015 and 2019[52]. Therefore, the coverage of floating kelp reported here should be updated as new information becomes available. Nonetheless, the compilation of maps presented here represents the most thorough global satellite assessment of kelp forest extent to date.

Our analysis uses the distribution of present surface canopy kelp, and it does not account for range contractions or expansions of kelp forests that are projected under climate scenarios[58,59]. Integrating future range shifts of kelp species and associated biodiversity under climate scenarios could guide the identification of climate-smart priority areas for kelp forest conservation[26]. Finally, the MPA dataset used here has some limitations regarding the quantification of protection. For example, it does not account for other human activities that MPAs can manage (e.g., mining, dredging) or indicators of management efficiency (e.g., budget, capacity, stage of establishment)[1] that need to be included to ensure MPAs are effectively protecting ecosystems[71]. Therefore, including such information will likely decrease the coverage of kelp forests within MPAs with high levels of effective protection because many lack effective governance, enforcement, or community involvement[72]. However, a comprehensive dataset of protection effectiveness is currently unavailable for all countries and MPAs (e.g., https://mpatlas.org/), and to date, ProtectedSeas[55] is the most complete database available to assess the level of restriction inside MPAs.

Kelp forests remain largely excluded from most international conservation policies[8,73], despite their enormous contribution to earth's biodiversity[12,13] and provisioning of ecosystem services[21]. Nations have an opportunity to harness, protect, and restore kelp forests[27,74], not only for their function as biogenic habitats and biodiversity hot spots[13], but also to support their role in carbon sequestration and mitigation of climate change[75]. In addition, kelp forests provide food and support the livelihoods of millions of people worldwide[13,21]. As part of efforts to protect 30% of the oceans by 2030[7], nations have an opportunity to explicitly include the representation of kelp forests in their national conservation policies[27]. Where nations share ecoregions, transboundary management and coordination may also be needed[26]. However, given the immediate and escalating threats posed by climate change[15,17,57] and other anthropogenic stressors, representation, though essential, may not be enough to safeguard the persistence of kelp forests. It is paramount that kelp forests are protected in each ecoregion through representative, adequate, and well-connected networks of climate-smart MPAs that consider additional climate adaptation strategies[26].

## Methods

### Mapping kelp forests

We compiled existing published regional and national datasets of surface-canopy forming kelp derived using remote sensing observations (Supplementary Table 1). We accessed quality-controlled estimates of kelp canopy derived from individually classified scenes observed by up to four Landsat sensors: Landsat 5 Thematic Mapper (1984–2011), Landsat 7 Enhanced Thematic Mapper+ (1999–present), Landsat 8 Operational Land Imager (2013–present), and Landsat 9 Operational Land Imager-2 (2021-present). The applicable Landsat observations have pixel resolutions of $30 \times 30$ m and repeat times of 16 days (8 days since 1999 in most years because two Landsat sensors were operational). Classification of floating kelp canopy was derived by applying a globally robust random forest classifier to individual Landsat scenes[76]. The compiled datasets include minor differences in methodologies and time periods, but they all cover approximately over 30 years (1984 onwards) (Supplementary Table 1). Kelp maps were created by compositing observations of kelp presence across this time series. The maps include most of the USA (California, Oregon, parts of Washington, and parts of Alaska) and all of Mexico, Peru, and Argentina (available at https://kelpwatch.org/)[76], most of the United Kingdom[77] (Malvinas/Falkland Islands), and most of Australia[78] (Tasmania). We included the Malvinas/Falkland Islands as part of the United Kingdom territory, although we acknowledge that Argentina has ongoing legal claims for their sovereignty.

For areas where the Landsat maps are not available, we supplemented the Landsat time series using available maps derived using empirical thresholding of Sentintel-2 satellite imagery. For the empirical Sentinel maps, kelp area was calculated using band-difference[52] or band-ratio[79] relationships and released for global and South African extents, respectively. The method for the empirical map applied to global waters averages all the available images from the Sentinel-2 satellite sensor from 26 June 2015 to 23 June 2019 to create a cloud-free mosaic. It then applies band-difference thresholds to identify pixels likely containing floating kelp canopy and a land mask using global digital elevation models (ALOS and SRTM), discarding topography with elevation >0 m. This dataset was validated across 14 in situ sites in South America that cover a variety of coastlines and ecoregions, and with existing data at 151 locations that cover four continents[52]. To ameliorate some potential detection caveats, we excluded pixels that fell within a 30-m buffer relative to the coastline because the global map does not distinguish between intertidal green algae and floating kelp forests and estuaries can also be a source of false positives. See Supplementary Table 1 for the coastlines used to apply the 30-m buffer. All kelp datasets were converted from coordinates to shapefiles with ArcMap10.8 using the World Geodetic System 1984 (WGS84). Our final floating kelp habitat map includes any pixel the satellite detected kelp in the time series and represents the known presence of floating kelp habitat in the timeseries. We included all observations to avoid arbitrarily choosing a period to map the distribution of floating kelp forests because these ecosystems are highly dynamic[76] and we were interested in detecting potential kelp habitat. For example, in some places kelps may be expanding their distribution (cold-edges)[58,59], while in others places kelps may be in alternative stable states dominated by urchin barrens[24] (degraded kelp ecosystems) or by more heat tolerant subcanopy kelp species (competing with floating kelps) near warm-edges[80]. These alternative stable states can shift, even after decades[81,82].

### Exposure of kelp forests to contemporary and future marine heatwaves

We estimated the expected threat of climate change to kelp forests by calculating historical and projected cumulative annual MHW intensities for each kelp pixel using a baseline climatology of 1983–2012. Marine heatwaves are periods during which temperature exceeds the

90[th] percentile of temperatures seasonally during a baseline period and last for at least five consecutive days[83]. To quantify the magnitude of present-day MHWs, we used the NOAA 0.25°-resolution Optimum Interpolation Sea-Surface Temperatures (OISST)[84] dataset (1982-present). Note that cumulative MHW intensities are an indicator of exposure[23,26], but they are not a measure of the vulnerability of species or ecosystems to MHWs

We also considered MHW characteristics using SST outputs from each of 11 Coupled Model Intercomparison Project Phase 6 (CMIP6; Supplementary Table 6) Earth System models (ESMs) re-gridded to 0.25° resolution using bilinear interpolation in CDO (Climate Data Operators). For each ESM, we selected the Historical run to represent the recent past (1983–2014), and selected three future (2015–2100) climate scenarios generated under the IPCC Shared Socio-Economic Pathways (SPPs)[53]: SSP1-2.6, SSP2-4.5, SSP5-8.5. SSP1-2.6 represents an optimistic scenario with a peak in radiative forcing at ~3 W m$^{-2}$ by 2100 (approximating a future with 1.8 °C of warming relative to the pre-industrial temperatures, in line with the Paris Agreement). SSP2-4.5 represents an intermediate mitigation scenario with radiative forcing stabilized at ~4.5 W m$^{-2}$ by 2100 (approximating implementation of current climate policies, resulting in 2.7 °C of warming by 2100). SSP5-8.5 represents an extreme counterfactual climate scenario with a continued rapid increase in greenhouse gas emissions resulting in radiative forcing reaching 8.5 W m$^{-2}$ (and 4.4 °C of warming) by 2100 and rising after that. We bias-corrected the SST dataset from each ESM relative to the corresponding ensemble mean of the Historical runs using the delta method (see[85]). This method ensures that projections for each ESM blend smoothly to the end of the Historical runs for the reference period 1983–2014. We then determined which grid cells overlayed with kelp forests, and when the kelp cell had no corresponding SST data for the ESM models (because ESMs have relatively coarse resolution), we filled the cell using the inverse-distance-weighted mean of surrounding cells.

We then used the R package *heatwaveR*[86] to estimate historical (1983–2020) and projected (2021–2100) cumulative annual MHW intensity (°C days) for each pixel. Note that although we used OISST data to quantify contemporary MHW intensities, we used corresponding data from each ESM's historical run for the period 1983–2014 when quantifying projected MHW intensities. This was necessary because using ESM data in the baseline period for projections instead of the OISST data ensures like-for-like comparisons (i.e., modeled data vs modeled data), avoiding issues associated with variation in inter-ESM skill in representing daily variability in SST. We used annual cumulative intensities because they are a good indicator of the exposure of kelp forests to warm anomalies[23,26]. We then estimated the median cumulative annual MHW intensity for each grid cell for the contemporary (2001–2020) period and across the 11 ESMs for the near- (2021–2040), mid- (2041–2060), and long-term (2081-2100) for each SSP and grid cell. Finally, we summarized trends in MHWs at the level of biogeographic realms and ecoregions[54] by conducting a spatial overlay (following the same approach as in the next sections).

### Marine protected areas: level of fishing restriction

We obtained the spatial boundaries of MPAs using two different sources of information for the countries that have surface-canopy forming kelp forests. First, we downloaded MPA boundaries from official country-level agencies (Supplementary Table 1). We undertook extensive searches to ensure that we used the most updated official information, as global datasets can be less comprehensive at the country-level. We then categorized each MPA based on the level of restrictions to extractive activities (recreational, subsistance, and commercial fishing). We used the Level of Fishing Protection (LFP) score obtained from ProtectedSeas[55] (https://protectedseas.net/). This database scores MPAs based on fishing restrictions on a scale of 1–5 scale (1 = Least restricted: no known restriction to the removal of

life, 2 = Less restricted: at least one species-or gear-specific restrictions apply, 3 = Moderately restricted: several species -or gear-specific restrictions apply, 4 = Heavily restricted: marine life removal is mostly prohibited with a few exceptions, 5 = Most restricted: marine life removal is prohibited). ProtectedSeas further divides the scores into categories: an LFP score of 1–2 is categorized as less protected, 3 as moderately protected, and 4–5 as highly protected areas, the most effective type of MPA[55]. Finally, we reviewed both country-level and ProtectedSeas datasets and, when needed, consulted country-level experts to ensure that all MPAs were included. We did not include other area-based measures not categorized as an MPA in the national dataset (e.g., fishery management areas). For a few MPAs (34 of 817) that had no LFP score, we reviewed existing information on restrictions on the removal of life and assigned a LFP score of less protected, moderately protected, or highly protected. We did not include other regulatory activities that MPAs can manage (e.g., mining, dredging, anchoring) or indicators of management efficiency (e.g., enforcement capacity, budget capacity, implementing management plan)[1] because such datasets are not comprehensively available for all countries.

### Global kelp distribution and protection

To estimate the amount of kelp within each level of protection, we performed a spatial intersection of MPA types (LFP classification; 817 spatial features) and the global kelp forest distribution (428,400 spatial features). Spatial intersection is a computationally expensive operation, so avoiding trivial calculations can significantly improve performance. We therefore developed and implemented a nested, parallelized, and hierarchical intersection algorithm. The approach is "nested" because spatial layers are split based on national jurisdiction before performing the spatial intersection. The approach is "parallelized" because the country-level intersection operations can be performed across parallel computer cores. Finally, the approach is "hierarchical" because, even within a country, not all kelp forests may lie within an MPA and not all MPAs may contain kelp. We first use a simple and less computationally expensive spatial join to identify kelp forests and MPAs that do not overlap with each other and exclude them from the expensive intersection calculation. Kelp forests excluded in this step are categorized as "not protected". Finally, we perform the spatial intersection between the kelp forests and MPAs that overlap. We then repeated this approach at the biogeographic realms and ecoregions as outlined by ref. 54. For all operations, we used unprojected coordinates (EPSG code 4326) that uses WGS84 datum and a spherical geometry engine (s2)[87] via the sf package[88] in R. Parallelization was done using the furrr and future[89] package in R. We validated geometries throughout the pipeline using st_make_valid in sf; any invalid geometries were removed.

Knowing the location and amount of kelp protected, we proceeded to calculate the total extent of kelp by country, biogeographic realm, and ecoregion, and by MPA category and LFP score. We also determined how much kelp was outside any protection. All spatial analyses were performed in R version 4.3.1 (2023-06-16)[90] using a x86_64-apple-darwin20 platform running macOS Ventura 13.4.1 and using the sf package v1.0[88,91] with GEOS 3.11.0, GDAL 3.5.3, and PROJ 9.1.0.

### Ecoregional marine heatwave threats under SSP2.4-5 and kelp representation

Our final analysis assessed the relationship between the threats posed by projected future MHWs to kelp forests and the amount (% area) protected in each ecoregion. We conducted this analysis at the ecoregional scale because, ideally, networks of MPAs should be established to protect the underlying biophysical processes that maintain species distribution and composition[26]. Areas with low values of projected future MHW intensities are potential climate refugia for kelp forests. For simplicity, our measure of threat is focused only on the average

cumulative MHW intensity under one SSP for each timeframe. We used SSP2-4.5 as an intermediate climate scenario that reflects less extreme outcomes and has been proposed to inform climate adaptation and policy[65,66]. Because the patterns of threat for each ecoregion are similar across time frames (i.e., magnitude is the largest difference across times), we focus in the main text on the mid-term and include results of the other times in the Supplementary information. We report results most conservatively for highly protected kelp, and then also for highly and moderately protected kelp combined. We did not include less protected MPAs in this analysis because this type of MPA provides minimal to no protection to marine ecosystems from extractive activities[1].

## Data availability

The remote-sensing kelp forest dataset is available at https://portal.edirepository.org/nis/mapbrowse?packageid=knb-lter-sbc.74.13, https://kelpwatch.org/map, and https://biogeoscienceslaboxford.users.earthengine.app/view/kelpforests. The marine protected area database is available at https://protectedseas.net/ upon request. The marine ecoregions of the world dataset is avaiable at https://databasin.org/datasets/3b6b12e7bcca419990c9081c0af254a2/. The compiled global floating kelp forest map and the processed marine protected area layer is available at Zenodo under the identifier https://doi.org/10.5281/zenodo.14796879. The raw data from figures and tables in this study are provided in the Source Data file. Source data are provided with this paper.

## Code availability

The codes used for this project is available at zenodo repository: https://doi.org/10.5281/zenodo.14796879[92].

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

## Acknowledgements

N.A.-D., K.C., and T.B. acknowledge support from the NASA Ocean Biology and Biogeochemistry program (80NSSC21K1429). N.A.-D. acknowledges support from the Estate Winifred Violet Scott (Australia) for a research grant. F.M., N.A.-D., and J.V.-D, acknowledge support from the National Science Foundation (2108566). N.A.-D., F.M., and K.C. acknowledge funding from the Lenfest Ocean Program (ID Number: 00036969). We are very thankful to the ProtectedSeas team for sharing the marine protected area dataset.

## Author contributions

N.A.-D. conceived the study with inputs from J.V.-D., D.S., F.M., and K.C. A.M.-S., T.B., H.H., L.D., C.B., and K.C. provided remote-sensing floating kelp forest datasets. J.S., and T.V. provided marine protected area datasets. N.A.-D., J.V.-D., and D.S. conducted analyses. N.A.-D. led reviewing nation-level marine protected area database with the support of A.M.-S., L.D., K.S., C.P., D.P., and C.L. N.A.-D. led the writing of the manuscript with the support of D.S., J.V.-D., F.M., and K.C. A.M.-S., T.B., C.B., M.C., L.D., H.H., C.L., C.P., D.P., K.S., J.S., and T.V. contributed to reviewing and editing of the manuscript.

## Competing interests

All other authors declare they have no competing interests.

## Additional information

¹Oceans Department, Hopkins Marine Station, Stanford University, Pacific Grove California, USA. ²Department of Geography, University of California Los Angeles, Los Angeles California, USA. ³Centre for Biodiversity Conservation, School of the Environment, University of Queensland, St. Lucia, QLD, Australia. ⁴MasKelp Foundation, Monterey California, USA. ⁵IUCN Species Survival Commission, Seaweed Specialist Group, Gland, Switzerland. ⁶Department of Environmental Science and Policy, Rosenstiel School of Marine, Atmospheric & Earth Science, University of Miami, Miami, FL, USA. ⁷Frost Institute of Data Science & Computing, University of Miami, Miami, FL, USA. ⁸Ocean Futures Research Cluster, School of Science, Technology and Engineering, University of the Sunshine Coast, Maroochydore, QLD, Australia. ⁹Department of Zoology, Centre for African Conservation Ecology, Nelson Mandela University, Gqeberha, South Africa. ¹⁰Department of Geography, University of Victoria, Victoria, British Columbia, Canada. ¹¹Department of Applied Ocean Physics and Engineering, Woods Hole Oceanographic Institution, Woods Hole, MA Massachusetts, USA. ¹²Institute of Marine and Antarctic Studies, University of Tasmania, Tasmania, Australia. ¹³University of Cape Town, Cape Town, South Africa. ¹⁴South African National Biodiversity Institute, Kirstenbosch Cape Town, South Africa. ¹⁵Institute for Coastal and Marine Research, Nelson Mandela University, Gqeberha, South Africa. ¹⁶Universidad Nacional de Córdoba, Facultad de Ciencias Exactas, Físicas y Naturales, Ecología Marina, Córdoba, Argentina. ¹⁷Consejo Nacional de Investigaciones Científicas y Técnicas (CONICET), Instituto de Diversidad y Ecología Animal (IDEA), Córdoba, Argentina. ¹⁸Fundación Por El Mar (PEM), Buenos Aires, Argentina. ¹⁹Sociedad, Ecología y Cultura, Lima, Perú. ²⁰ProtectedSeas, Anthropocene Institute, Palo Alto California, USA. ²¹Stanford Center for Ocean Solutions, Stanford University, Pacific Grove California, USA. ✉e-mail: n.arafehdalmau@uq.net.au

