## [Peer Review File · Nature Communications]

Global floating kelp forests have limited protection despite intensifying marine heatwave threats

Corresponding Author: Dr Nur Arafeh-Dalmau

Version 0:

Reviewer comments:

Reviewer #1

(Remarks to the Author)

This review was jointly done by an ECR and mid-career researcher. This study combined maps of floating kelp forests, environmental data on future marine heatwaves under different climate scenarios, and MPA spatial data to identify the amount of kelp forests most vulnerable to MHWs and areas that are currently protected. This is important information in light of the 30X30 goals. Overall, we enjoyed reading the paper and found the use of maps and figures informative and the code and data availability suitable. The topic of kelp and MHWs is timely and the research topic should be of interest to a broad readership. However, we do have some concerns with the conceptual underpinnings of the research question, and the presentation of the results, which should be addressed.

Main comments

The authors emphasize the use of MPAs as the “most effective” conservation measure for safeguarding kelp, stating they can provide “climate resilience” or “enhance their adaptability”. However, there is very limited evidence that MPAs increase the resilience of kelp forests to warming temperatures or increase how they can adapt to extreme events. The only documented case of MPAs buffering MHW impacts is the example given on line 270, which may be the exception and not the rule. MPAs are not always effective at mitigating MHW impacts because one of the main mechanisms which MHWs impact kelp forests is through direct mortality of kelp by surpassing temperature thresholds (Filbee-Dexter et al. 2021, Thomsen et al. 2021, Smale 2020). MPAs can do little to nothing against this lethal stress. MPAs may help kelps better survive non-lethal warming temperatures by reducing other stressors, yet in situ evidence of this mechanism is scarce. This was overviewed by Bates et al.’s ‘protection paradox’ study. In general, MPAs can potentially prevent kelp loss in areas where the MHW causes kelp loss due to indirect drivers that can be managed to some extent by top-down control. For example, in some regions where they are overgrazed by sea urchins (Tasmania, North California). Yet, even in areas where this link is best established such as New Zealand in the Leigh Reserve, there is no overall relationship between kelp cover and protection level at broader scales across multiple reserves in NZL (and this does not even include MHW stress). Further, MPAs do not help rebuild all urchin predators. Kelp forests in the North Pacific are impacted by otter loss, and MPAs are not a typical tool to rebuild otter populations. Same can be said for large groundfish in the North Atlantic. The limitation of MPAs against this heatwave driver should be better explored and addressed much earlier in the study.

The study implies that MHWs impact equally along a kelp species range. Yet, MHW impacts are likely more severe for kelp forests at the canopy species’ warm range edge, compared to cool range edges. This is mentioned on line 260 as a bit of a throwaway discussion comment - but this is essential to determine vulnerability. The presentation of the vulnerability results treat warm and cool range edges as if they are equally vulnerable to MHWs. For example, the study identifies the Bering Sea, the Gulf of Alaska, the North American Fjordlands and the Puget Trough as vulnerable areas for kelp forests. Yet, the kelp species in these Arctic and sub-Arctic areas are predominantly at the cool range edge of their thermal tolerance. This means that increased sea temperatures of a few degrees from a MHW likely will not cause direct mortality and could even make temperatures temporarily more suitable for these species. It is more likely that kelps in California and northern Chile are more vulnerable.

The authors discuss that their analysis excludes many kelp species (line 288) and only includes surface-forming kelps. The fact that this paper shows the distribution and lack of protection for only three species of kelp out of >100 Laminarian kelp should be emphasised. Kelp forests made by subsurface species will overlap this distribution and are found along 30% of the world's coasts. This is not a global analysis - but an analysis of three widespread canopy kelp species. The title of this paper therefore cannot include the word 'global' as it is misleading.

Finally, the paper often refers to the amount of protection in percentages, it would be more effective to also mention the area/scale of unprotected kelp forests here. The authors show kelp area per ecoregion, but that can be better communicated in text.

See more specific comments/ suggestions by line.

Line 55: Replace "for all forests" with "across all regions".

Line 55: The use of the word 'emerge' seems misguided here, we suggest the end of this sentence could read "...some kelp forests in the southern hemisphere may be protected within climate refugia".

Line 71: Add "and extreme events" after "threats associated with climate change".

Line 84: Change "trophic-level" to "trophic level organisms".

Line 87: For scale/ context, the authors should consider providing the average canopy extent for each of these species in a known area.

Line 89: Reference needed after "threats posed by MHWs".

Line 90: Reference needed after "protection status".

Line 90: Identifying climate refugia alone is not a climate adaptation strategy. Instead, that information needs to be considered or applied in the context of conservation management as indicated below, there is need to reframe this sentence.

Line 93: After "can persist" change the sentence to read "...and the resilience of other kelp forests may be enhanced, depending on local and regional connectivity and species life-history traits, by maintaining a source of recovery for impacted kelp habitats."

Line 117: Change "while" to "while the remaining".

Line 118: Change "forests" to "forests, combined".

Line 119: Add ", overall," after "highest latitudes" to keep consistent with the following.

Lines 121-122: Remove "at their warm distribution limit" and change "to their highest latitudes in" to "southwards to their cool-distribution limit in".

Line 123: Change "include" to "are located in".

Line 144: Add "realms" after "Pacific".

Lines 150-151: Change the sentence to read "Specifically, projections suggest a latitudinal pattern of increasing exposure to future MHWs from lower to higher latitudes in the northern hemisphere, whereas..."

Figure 1: Why does the Arctic region (green) have a large coloured portion that does not match the corresponding black line showing kelp distribution?

Line 174: Remove the plural from "MPAs".

Line 181: So even though France meets the current 30% target, does this mean that all their kelp forests are located on remote islands and there are no protection measures on the mainland?

Line 183: Here, it would be more effective to also mention the scale of continental kelp forests that are unprotected in these countries since it will be a very large area. Only reporting on percentages takes away from this point.

Line 193: Add "Overall" before "only".

Line 281: Change "threated" to "threatened".

Line 240: Change "country" to "nation" to keep consistent with the above.

Line 262: provide some reasons or justification why it is challenging.

Lines 267-269: This point needs to be made clearer earlier in the paper for example, around lines 80-84.

Line 271: After "less biodiverse ecosystems" outline what species these systems were previously dominated by, compared to how they look now after the MHW.

Line 272: Remove "MPAs have prevented" and "have provided". Replace "collapse" with "collapse was prevented".

Line 273: add was 'shown'

Line 275: Replace "other" with "supplementary".

Lines 278-279: Consider "other anthropogenic impacts" both in the ocean and on land. What manageable stressors are/ can be identified? This calls on local conservation managers to have a solid understanding of compounding stressors, aside from warming and extreme events.

Line 288: The point that this analysis only includes surface canopy kelp is underemphasised throughout the manuscript. While we understand the complexities and limitations of gathering reliable global datasets, could the authors expand on the extent of the surface-canopy kelp distribution relative to known sub-canopy kelp distributions for key areas, for example, Australia, New Zealand, Chile, South Africa, USA etc?

Line 329: Replace "secure" with "safeguard".

Line 388: After "by 2100" outline what the relative warming would be, as above.

Line 414: What are some examples of these extractive activities relative to each of the levels?

Lines 422-423: What are the country-level criteria for fishing restrictions? The authors also mention that they do not include other types of spatial closures and area-based measures that are not MPAs, by assumption this varies from country to country therefore. Some insight should be provided, for example, does this include cultural or seasonal-based closures? MPAs are not all created equal, and this is an important point to be clear on.

(Remarks on code availability)

Reviewer #2

(Remarks to the Author)

(Remarks on code availability)

Reviewer #3

(Remarks to the Author)

This work focuses on exposure of global floating kelp forests to marine heatwaves and analyze the spatial distribution of kelp forests with respect to the marine protected areas. Although interesting results are presented, major concerns exist, especially in terms of the soundness of methodology and results.

Result completeness. As only the floating (surface canopy) kelp forests can be detected from satellite image, and only 3 of the largest kelp species form floating canopies out of over 120 kelp species, it is not accurate to say that the presented results from satellite observations here are "global kelp forests", as subsurface kelp forests have not been reasonably accounted for.

Result reliability. Classification of floating kelp canopy was derived by applying a globally robust random forest classifier, and the global map was only validated using ground observations at 14 sites in South America. It is largely unknown how the classifier works elsewhere and if the classification accuracy is good enough to support the analysis presented here.

Results consistency. As Sentinel-2 satellite imagery (2015-2019) was adopted to supplement Landsat observations (1984-98 present), how to guarantee the consistency (or comparability) between results from different satellite data, especially in terms of different band configuration and algorithms used to kelp forest?

Conclusion soundness. A key conclusion in Abstract is that "...exposure to projected marine heatwaves will increase ~8 times compared to contemporary (2001-2020) exposure for intermediate climate scenarios". But I didn't find the corresponding statement in the main text, especially in terms of "the 8 times". How was it defined and calculated?

Title is not correct and logical. The fact is that intensifying marine heatwaves and anthropogenic activities may threaten global kelp forests, but kelp forests cannot be threatened by protection behavior even if the protection is limited.

In captions of Figure 2 it read SSP-1.26, SSP-2.45, SSP-5.85, whereas in the figure it read different as SSP1-2.6, SSP2-4.5, SSP5-8.5. This issue also exists in the methods parts.

(Remarks on code availability)

Reviewer #4

(Remarks to the Author)

This ms uses remotely sensed data to quantify the distribution of canopy-forming kelp across the world's oceans before overlaying that with MPA designations under different levels of fisheries restrictions. They then calculate cumulative MHW statistics for contemporary times and under different climate forcing scenarios for near and far timescales. They show that kelp regional differences in the susceptibility to MHWs into the future, but also that because the vast majority of kelp forests are not highly protected there will be reduced resilience to MHWs. Given the important role that kelp forests play in supporting elevated levels of diversity as well as providing extensive ecosystem services the impacts of increased MHWs and the lack of protection will have wide ranging consequences. I think this is a timely paper which brings together extensive datasets to undertake the analyses presented. I think that in places the text needs to be toned down and caveated (see specific comments below), but the conclusions are generally sound and the call for better protection for kelp forests is welcomed.

Specific comments

Line 50 I don't think there is evidence that climate change poses the greatest risk and your analysis suggests some area may be more at risk from others. I would rephrase.

Line 51 I would rephrase the last part of this sentence as there is quite a lot of literature on the future threats to kelp and at least in a regional context and understanding of the current conservation status (or lack thereof). Suggest rephrasing

Abstract – Suggest adding floating to all discussion of kelp in the abstract as it suggests as written that the outputs are extrapolated to all kelps, which isn't the case and also not possible with this analysis.

Line 58 “the most effective conservation measure for providing climate resilience” I find this phrase a little difficult throughout. Yes highly restrictive marine protected areas that stop habitat loss and ensure intact predators in kelp systems controlled by grazers are likely to increase climate resilience, but if the chief driver of kelp loss is say invasive species/ eutrophication (as it is in some parts of the world) this type of MPA would add little to make kelp forests for resilient to climate change. I would suggest using more nuanced language.

Lines 71-72 I suggest rephrasing this passage as we generally have a good global understanding of kelp forests distributions and an understanding of responses to climate change and MHWs from in-situ studies and experiments. I agree at regional scales kelp forest extent can be much more poorly resolved.

Line 74 Productivity is more similar to broadleaf forests than rainforests or coral reefs. See Pessarrodona et al (2022) Science Advances eabn2465 – Fig 1a

Line 76-77 I am not convinced by this line “Kelp forests are among the marine ecosystems at greatest risk from MHWs”. Surely this depends on where the MHW occurs in the species range and I would suggest that there is equal evidence that other species and ecosystems (e.g. coral reefs) are equally or more at risk. I suggest toning the language.
Line 78 I am not suggesting that MHWs haven't contributed to the loss of kelp in Tasmania, but the key driver is expansion of urchins (as a result of decadal scale warming) and over fishing of large rock lobster. Suggest altering the language or removing Tasmania as an example of MHW driven kelp loss.

Line 106 I know that you have defined above that by kelp forests you mean canopy-forming forests, but in places throughout the ms I think it would be useful to emphasise that the ms is only referring to canopy-forming forests and I think this is one of those instances.

Line 136 Should be Fib 1b and 2 referred to here

Line 145 Suggest adding temperate South Africa here as they look very similar to temperate South America.

Figure 2b I suggest using the same labelling system for the SSP as in the legend i.e. the – after SSP. Also I think that the SSPs are incorrectly labelled. At present SSP – 1.26 has the most extreme cumulative MHWs. Finally I suggest changing the colouring for contemporary and (currently labelled) SSP – 5.85.

Line 173 Suggest emphasising floating kelp forests here

Line 186 It might be worth some discussion in the ms about whether these are real or paper HPMPAs, particularly in the Southern Ocean, given their isolation (maybe some discussion of this in the section beginning line 304). Also what does this 99% protection represent in terms of global kelp? If 99% of small amount of kelp forest this doesn't really mean anything. Might be worth including this information for context.

Figure 3 may be worth also including the island/islands that are protectorates of France rather than just stating France. Maybe consider the same for UK protectorates although these do come out in the text.

Using level of fishing protection is only a good measure of protection if grazing is a key pressure and more intact fish stocks reduce this or that destructive fishing gears are removed which reduces habitat loss. Where grazing by urchins may not be a key driver these HPMPAs may not increase resilience. Is there confidence that grazing is a key driver across all these regions? It isn't in all kelp forests and if there isn't confidence then I think that this needs caveating in the ms – maybe in the methods, but potentially also in the main ms (maybe around lines 245-246?).

The identification of refugia is made in the abstract and line 277, but I am somewhat sceptical that areas of lower MHW impact will be able to act as a refugia given the distance between the most and least impacted areas. The identified refugia are in the Southern Ocean while the most impacted are in the Pacific Arctic. *Macrocystis* has greater dispersal ability than many kelps, but this seems to be stretching the resupply from refugia further than perhaps it should go. I believe this needs some discussion and toning down.

Lines 292-295 It is stated that there will be some overlap with floating and subsurface kelp and therefore the outcomes may be similar for kelp in these regions. I agree, but I think it is worth stating that whole ocean basins e.g. NE Pacific and N Atlantic with kelp present are not represented at all as they don't possess floating kelp.

Line 366 It isn't clear to me what the filters and masks are in Supplementary Table 1

Line 368-369 Given the evidence for regional losses of kelp over the last 20-30 years I am a little sceptical of using Landsat images back to 1984. What if all the detections were early in the dataset and there have none in the later years? Was the potential for range reductions in kelp forests over the time-series considered? If it wasn't I suggest that some sort of temporal analysis of this should take place before using data back to 1984.

Line 373 I suggest that you make the length and years of the baseline period clear here. It is mentioned below, but it would be best to mention when first stated.

Line 398 It isn't clear to me why a different base line would be used for the reference period and the climatology. When not use the same period?

Lines 416-419 I think it would be helpful what the different categories mean in terms of restrictions and LFP.

(Remarks on code availability)

Version 1:

Reviewer comments:

Reviewer #1

(Remarks to the Author)
the authors have addressed all comments suitably

(Remarks on code availability)

Reviewer #2

(Remarks to the Author)
I co-reviewed this manuscript with one of the reviewers who provided the listed reports. This is part of the Nature Communications initiative to facilitate training in peer review and to provide appropriate recognition for Early Career Researchers who co-review manuscripts.

(Remarks on code availability)

Reviewer #3

(Remarks to the Author)
All my concerns have been addressed.

(Remarks on code availability)
N/A

RESPONSE TO REVIEWERS

Reviewer 1 and 2

General Comment Reviewer 1

This review was jointly done by an ECR and mid-career researcher. This study combined maps of floating kelp forests, environmental data on future marine heatwaves under different climate scenarios, and MPA spatial data to identify the amount of kelp forests most vulnerable to MHWs and areas that are currently protected. This is important information in light of the 30X30 goals. Overall, we enjoyed reading the paper and found the use of maps and figures informative and the code and data availability suitable. The topic of kelp and MHWs is timely and the research topic should be of interest to a broad readership. However, we do have some concerns with the conceptual underpinnings of the research question, and the presentation of the results, which should be addressed.

General Comment Reviewer 2

Response General Comment Reviewer 1 and 2

Thank you so much for your valuable review, which has improved our manuscript.

Response General Comment

We appreciate the reviewer's encouraging comments and that they enjoyed reading our manuscript and found our work important, timely and relevant.

Main comment 1

The authors emphasize the use of MPAs as the “most effective” conservation measure for safeguarding kelp, stating they can provide “climate resilience” or “enhance their adaptability”. However, there is very limited evidence that MPAs increase the resilience of kelp forests to warming temperatures or increase how they can adapt to extreme events. The only documented case of MPAs buffering MHW impacts is the example given on line 270, which may be the exception and not the rule. MPAs are not always effective at mitigating MHW impacts because one of the main mechanisms which MHWs impact kelp forests is through direct mortality of kelp by surpassing temperature thresholds (Filbee-Dexter et al. 2021, Thomsen et al. 2021, Smale 2020). MPAs can do little to nothing against this lethal stress. MPAs may help kelps better survive non-lethal warming temperatures by reducing other stressors, yet in situ evidence of this mechanism is scarce. This was overviewed by Bates et al.'s ‘protection paradox’ study. In general, MPAs can potentially prevent kelp loss in areas where the MHW causes kelp loss due to indirect drivers that can be managed to some extent by top-down control. For example, in some regions where they are overgrazed by sea urchins (Tasmania, North California). Yet, even in areas where this link is best established such as New Zealand in the Leigh Reserve, there is

no overall relationship between kelp cover and protection level at broader scales across multiple reserves in NZL (and this does not even include MHW stress). Further, MPAs do not help rebuild all urchin predators. Kelp forests in the North Pacific are impacted by otter loss, and MPAs are not a typical tool to rebuild otter populations. Same can be said for large groundfish in the North Atlantic. The limitation of MPAs against this heatwave driver should be better explored and addressed much earlier in the study.

Response main comment 1

Done. We thank the reviewers for commenting on our statement that MPAs are the “most effective” conservation measure to adapt to climate change. We have modified the abstract to ensure we clarify that highly restrictive MPAs are the most effective MPAs.

Orange text is added/modified text, applies throughout.

Now Lines 59-61: “less than 3% of global kelp forests are currently within highly restrictive marine protected areas (MPAs), the most effective MPAs for protecting biodiversity.”

Done: We apologize if our framing was missing further detail. We agree (discussed in paragraph 3 of the discussion) with the reviewers that MPAs have limited capacity to directly mitigate climate change impacts that surpass physiological thresholds. We have added text to clarify when MPAs may promote resilience. For example, MPAs can support the recovery of key species for ecosystem functioning and stability, which can enhance the capacity of ecosystems to resist and recover from MHWs by reducing other stressors besides temperature. To address the reviewer’s comments, in the introduction, we have pointed out limitations and expanded the benefits that MPAs can provide to kelp forest ecosystems, beyond climate resilience. We also reference studies reporting evidence that the trophic cascade mechanism can promote resilience when urchin predators are protected from fishing in MPAs, from different geographies (Filbee-Dexter et al. 2024, Kumagai et al. 2024, Ling et al. 2009, Peleg et al. 2023). In addition, we have added more examples that empirically found that MPAs facilitate resilience among kelp forest ecosystems through other mechanisms (Micheli et al. 2012, Munguia-Vega et al. 2015, Ziegler et al. 2023, Benedetti-Cecchi et al. 2024, Olguin-Jacobson et al. 2024) and provided details of the only study that comprehensively evaluated whether a network of MPAs provides resilience to MHWs through trophic cascades (Kumagai et al. 2024). This additional review references and provide a more comprehensive overview of this topic and we thank the reviewers for raising this issue.

Now Lines 86-101 in the Introduction: “While MPAs cannot directly counter the impacts of climate change that can surpass a species’ physiological tolerance²⁷, MPAs can mitigate non-climatic stressors like overfishing and habitat destruction, which can enhance ecosystem resilience²⁷, supporting ecological functioning and providing societal benefits²⁸⁻³⁰.

Well-managed and highly restrictive MPAs —no-take marine reserves, where all fishing activities are prohibited— are the most effective type of MPA for rebuilding fished

populations²⁸, supporting the stability of kelp forest ecosystems³¹, and, in some documented cases, providing their resilience to MHW impacts³²⁻³⁵. For example, in regions where urchin predators are protected from fishing and where trophic cascades are a driver of food-web dynamics, MPAs can facilitate the recovery of higher-trophic-level organisms, which helps control kelp grazer populations and prevent overgrazing of kelp³⁶⁻³⁸. This mechanism has been found to support resistance to and recovery from MHWs for kelp forests in a network of 39 MPAs in southern California³⁸. In addition, MPAs can facilitate climate resilience for kelp forest ecosystems through other mechanisms^{32-34,39, 40}. For example, a recent global analysis found that fish communities were more stable in the face of MHWs in highly restrictive MPAs than unprotected sites³³, and abalone populations in two MPAs in Mexico were more resilient to hypoxia and MHWs because of their larger body size and greater reproductive output^{33,40}.”

We want to clarify that our work evaluates the protection of kelp forests as a habitat that structures an ecosystem and not as a single species. This aligns with general conservation principles of habitat representation as biodiversity indicators (Margules & Pressley 2000), which links with the global targets of protecting 30% of habitats by 2030. This is clearly stated in the opening paragraph (Lines 69-70), and in the discussion (Lines 276-281), where, when discussing threats and protection, we refer to “kelp forest ecosystems”. We have added more detail in other sections to avoid confusion, including the word “representation” in the title. This is addressed in following response (see answer Main Comment 3, Page 5).

Title

“Global floating kelp forests have limited representation in marine protected areas despite rising threats from marine heatwaves”

Introduction

Now Lines 125-126: “global map of floating kelp forest habitats (henceforth “kelp forests”)”

Results

Now Lines 143: “We found floating kelp forest habitats in ...”

Now Lines 156: “The exposure of kelp forest habitats to ...”

Now Lines 207: “Globally, more than 33.1% of floating kelp forest habitats are”

Discussion

Now Lines 274: “global map of the protection status of floating kelp forest habitats”

Main comment 2

The study implies that MHWs impact equally along a kelp species range. Yet, MHW impacts are likely more severe for kelp forests at the canopy species’ warm range edge, compared to cool range edges. This is mentioned on line 260 as a bit of a throwaway discussion comment - but this is essential to determine vulnerability. The presentation of the vulnerability results treat

warm and cool range edges as if they are equally vulnerable to MHWs. For example, the study identifies the Bering Sea, the Gulf of Alaska, the North American Fjordlands and the Puget Trough as vulnerable areas for kelp forests. Yet, the kelp species in these Arctic and sub-Arctic areas are predominantly at the cool range edge of their thermal tolerance. This means that increased sea temperatures of a few degrees from a MHW likely will not cause direct mortality and could even make temperatures temporarily more suitable for these species. It is more likely that kelps in California and northern Chile are more vulnerable.

Response Main comment 2

Done. We agree with the reviewer's comment that MHWs will not impact equally along a species range. We thank the reviewer for this comment, which allowed us to expand the text to clarify these important points further. We want to clarify that our work does not assess the vulnerability of kelps to future MHWs but exposure, which is a measure of potential threat rather than species vulnerability. Assessing vulnerability would require a different study that considers the susceptibility of kelp populations to MHW threats considering biotic and abiotic factors that drive species' local adaptation and range shifts (see recent review in Nature Reviews Earth & Environment, Lawler et al. 2024). This is important because, for example, studies found that some of the most resilient kelp forests in the California Current following MHWs are in the species warm-distribution limit in Baja California (see Cavanaugh et al. 2019 and Bell et al. 2023). These kelp forests have adapted to higher temperatures and large anomalies (temperatures peak to >25°C during MHWs). We have modified the discussion paragraph to better frame and explain the important points raised by the reviewer.

Now Lines 296-306 in the Discussion: Note that our study assessed the exposure of kelp forests to projected MHWs, not their vulnerability to MHWs. While kelp forests near their current warm distribution limit will likely be the most affected and therefore subject to range contractions^{15,56,59,60}, populations living further from their thermal tolerance limit may be less threatened, and novel climates may favor kelp expansion^{56,57}. However, cold-range populations subject to extreme MHWs could also be threatened, as they may be less adapted to extreme temperatures⁶¹. Moreover, the persistence of kelp forest ecosystems is not driven solely by available substratum and suitable temperatures but also by biotic interactions⁶², and not all components of these ecosystems will necessarily respond in the same way. Predicting whether MPAs can provide resilience to kelp forest ecosystems under such extreme and persistent changes is therefore challenging on multiple fronts.

We have also reduced specific discussion of the cold-edge populations to ensure that comments are more general for North America (the continent with the highest projected MHW threat), including for warm-edge distribution limits.

Now Lines 319-323 in the Discussion: "Because the magnitude of future warming may cause the loss of kelp forests in some regions, MPAs will likely not be enough on their own to support the persistence of kelp forests. In these cases, supplementary climate-

adaptation strategies will be necessary, particularly for areas of high exposure to future MHWs, such as **regions in North America and especially areas near-warm distribution limits**".

In addition, to avoid confusion, we have added the following detail regarding our analyses.

Now Line 436-438 in the methods: "Note that cumulative MHW intensities are an indicator of exposure^{22,25}, but they are not a measure of the vulnerability of species or ecosystems to MHWs."

Main comment 3

The authors discuss that their analysis excludes many kelp species (line 288) and only includes surface-forming kelps. The fact that this paper shows the distribution and lack of protection for only three species of kelp out of >100 Laminarian kelp should be emphasised. Kelp forests made by subsurface species will overlap this distribution and are found along 30% of the world's coasts. This is not a global analysis - but an analysis of three widespread canopy kelp species. The title of this paper therefore cannot include the word 'global' as it is misleading.

Response Main comment 3

Done. We agree with the reviewer that our work focus on the detection of floating kelp forests globally, which are composed of a few species. However, these species are the only ones that can currently be comprehensively mapped (i.e., at large spatial and temporal scales), and they are globally distributed across all continents. Floating kelps co-exist with many other kelp species and structure one of Earth's most biodiverse ecosystems. Therefore, maps of the presence of floating kelp forest canopies is an excellent indicator of the presence of floating kelp forest ecosystems that include many other kelp species, and hundreds to thousands of species of seaweeds, fish, and invertebrates, none of which can be mapped remotely.

We thank the reviewer's comments, which allowed us to clarify that our work is not single-species focused but ecosystem-focused (as mentioned in early comments) and better explain why floating kelp forest maps are excellent indicators of the presence of kelp forest ecosystems and associated species. We also changed the title to clarify that we are referring to globally distributed floating kelp.

Now Lines 105-118 in the Introduction: "Recent advances in satellite imaging of surface-canopy-forming kelp species provide an opportunity to map the distribution of kelp forest **habitats³¹, quantify the threats posed by MHWs, and assess their protection status. **Floating kelp forests are a globally distributed foundation species that co-exist with other sub-canopy kelp species that structure one of Earth's most biodiverse ecosystems**¹². **These forests can cover thousands of hectares (e.g., 28,500 hectares in the Southern California Bight ecoregion⁴²) and sustain hundreds to thousands of species, some of which are economically and culturally significant. For example, in the Channel Islands in****

Southern California, studies found 716 species associated with giant kelp forest ecosystems⁴³ and in Patagonia, Chile and Argentina, similar studies found between 150-250 species⁴⁴⁻⁴⁶. In addition, in the northeast Pacific Ocean, 17 species of sub-canopy kelp coexist with floating kelp forests (www.algaebase.org/) from Alaska to Baja California, Mexico. Since remote sensing is the only available method to detect kelp forest ecosystems comprehensively (i.e., using standardized methods at large temporal and spatial scales), maps of floating kelps are good indicators of the broader ecosystem and associated biodiversity.

Main comment 4

Finally, the paper often refers to the amount of protection in percentages, it would be more effective to also mention the area/ scale of unprotected kelp forests here. The authors show kelp area per ecoregion, but that can be better communicated in text.

Response Main comment 4

Done. We have added some more detail of the coverage of kelp in the text.

Results:

Now Lines 210-211: “the Southern Ocean realm (24,319.8 ha), and when excluding these areas, only 2.8% (5,870.9 ha) of the global kelp forests are highly protected...”

Now Lines 215-219: “However, this protection is in their overseas territories in remote islands for all of France (23,007.1 ha, there is no floating kelp forests in mainland France) and much of New Zealand (145.1 ha), South Africa (285.1 ha), and Australia (78.7 ha). Australia has only 2.7% (23.0 ha), New Zealand 2.0% (14.1 ha), and South Africa 8.9% (400.6 ha) of their continental kelp forests highly protected.”

Specific comments

See more specific comments/ suggestions by line.

Line 55: Replace “for all forests” with “across all regions”.

Line 55: The use of the word 'emerge' seems misguided here, we suggest the end of this sentence could read "...some kelp forests in the southern hemisphere may be protected within climate refugia".

Line 71: Add “and extreme events” after “threats associated with climate change”.

Line 84: Change “trophic-level” to “trophic level organisms”.

Done

Line 87: For scale/ context, the authors should consider providing the average canopy extent for each of these species in a known area.

Done. Although the exact extent is not available because current satellite methods don't allow distinguishing among species. We have provided one example that is published for giant kelp in the Southern California Bight ecoregion, as an example.

Now Lines 109-110 in Introduction: “These forests can cover thousands of hectares (e.g., 28,500 hectares in the Southern California Bight ecoregion⁴²)”

Line 89: Reference needed after “threats posed by MHWs”.

Line 90: Reference needed after “protection status”.

Done. Moved (Cavanaugh et al. 2021) after “protection status”. This reference is a review of the use of remote sensing for floating kelp forest mapping, threats and management .

Now Line 107.

Line 90: Identifying climate refugia alone is not a climate adaptation strategy. Instead, that information needs to be considered or applied in the context of conservation management as indicated below, there is need to reframe this sentence.

Done. Now Line 119: “strategies such as identifying and protecting climate refugia...”

Line 93: After “can persist” change the sentence to read “...and the resilience of other kelp forests may be enhanced, depending on local and regional connectivity and species life-history traits, by maintaining a source of recovery for impacted kelp habitats.”

Done. Now Lines 121-123: “biodiversity can persist⁴⁷ and may enhance the resilience of other kelp forests, depending on local and regional connectivity and live history-traits...”

Line 117: Change “while” to “while the remaining”.

Done

Line 118: Change “forests” to “forests, combined”.

Done. Now Lines 147-148: “while the remaining 17 ecoregions combined account for only 1% of the distribution of kelp forests (Supplementary Fig. 1).”

Line 119: Add “, overall,” after “highest latitudes” to keep consistent with the following.

Lines 121-122: Remove “at their warm distribution limit” and change “to their highest latitudes in” to “southwards to their cool-distribution limit in”.

Line 123: Change “include” to “are located in”.

Line 144: Add “realms” after “Pacific”.

Lines 150-151: Change the sentence to read “Specifically, projections suggest a latitudinal pattern of increasing exposure to future MHWs from lower to higher latitudes in the northern hemisphere, whereas...”

Done

Figure 1: Why does the Arctic region (green) have a large coloured portion that does not match the corresponding black line showing kelp distribution?

There is no detected floating kelp in parts of that region (like others, see South Africa)

Line 174: Remove the plural from “MPAs”.

Done

Line 181: So even though France meets the current 30% target, does this mean that all their kelp forests are located on remote islands and there are no protection measures on the mainland?

Done. These means that all their floating kelp forest are protected and in remote islands. We have further clarified in Results, Now Lines 215-217: “However, this protection is in their overseas territories in remote islands for all of France (23,007.1 ha, there is no floating kelp forests in mainland France)”

Line 183: Here, it would be more effective to also mention the scale of continental kelp forests that are unprotected in these countries since it will be a very large area. Only reporting on percentages takes away from this point.

Done. See previous response for Main Comment 4 (Page 6).

Line 193: Add “Overall” before “only”.

We use the word overall in the previous sentence, it seems repetitive to use it again.

Line 281: Change “threatened” to “threatened”.

Done. Although we think you mean line 218?

Line 240: Change “country” to “nation” to keep consistent with the above.

Done

Line 262: provide some reasons or justification why it is challenging.:

Done. See previous response to main comment 2 in page 4. Now Lines 301-306.

Lines 267-269: This point needs to be made clearer earlier in the paper for example, around lines 80-84.

Done. Addressed in previous comments (Page 2). Now Lines 86-101 in the Introduction.

Line 271: After “less biodiverse ecosystems” outline what species these systems were previously dominated by, compared to how they look now after the MHW.

Done. Now Lines 315-317: **Studies in the Channel islands have shown that urchin barrens reduced sessile invertebrate diversity by 40 % and almost completely removed the canopy fish assemblage⁴³.**

Line 272: Remove “MPAs have prevented” and “have provided”. Replace “collapse” with “collapse was prevented”.

Done. We have changed the sentence. Now Lines 317-319:

However, highly protected MPAs have **promoted the resilience of kelp forests following MHWs** by facilitating recovery of overfished predators that control urchin populations^{29,30}

Line 273: add was 'shown'

The reviewer does not detail where we need to add "shown" in the sentence, and from our reading it is not clear where this would fit.

Line 275: Replace "other" with "supplementary".

Done.

Lines 278-279: Consider "other anthropogenic impacts" both in the ocean and on land. What manageable stressors are/ can be identified? This calls on local conservation managers to have a solid understanding of compounding stressors, aside from warming and extreme events.

Done. Thank you for the thought; we have added in parenthesis an example.

Now Lines 325-326: "and managing other anthropogenic impacts (e.g., land-based pollution) not mitigated by MPAs".

Line 288: The point that this analysis only includes surface canopy kelp is underemphasised throughout the manuscript. While we understand the complexities and limitations of gathering reliable global datasets, could the authors expand on the extent of the surface-canopy kelp distribution relative to known sub-canopy kelp distributions for key areas, for example, Australia, New Zealand, Chile, South Africa, USA etc?

Done. We have expended both the introduction and discussion and specified that the floating kelp maps are indicators of the distribution of kelp forest ecosystems and provided cases where our maps may underrepresent kelp forest ecosystems. See previous response Main Comment 3, page 5.

And Discussion Now Lines 335-349: "It is important to note that our analysis maps floating kelp forests, thus our method will not detect other kelp forest ecosystems when they do not co-occur with floating kelps. There are > 120 laminarian kelp species (many are intertidal), of which three of the largest kelp species form extensive floating canopies that can be detected by remote sensing, including the globally distributed *M. pyrifera*. Our estimates likely represent overall kelp forest distribution and ecosystem protection in regions where floating kelps co-exist with other sub-canopy kelp species (e.g., the west coast of North America, South America, and remote southern hemisphere islands, among others). However, some other nations and regions not included here have extensive subsurface kelp forest ecosystems which do not co-occur with floating kelp species (e.g., the east coast of North America, northern Europe, parts of Australia, among others). Given the limitations in detecting subsurface canopy kelp forests, they are likely less-well represented here than those detected by remote sensing. This is a substantial gap for kelp conservation and an avenue for novel technologies and research⁶⁷ to address associated needs for these kelp ecosystems that do not overlap with floating kelp, which also support diverse and productive ecosystems^{13,68} and human livelihoods²¹."

Line 329: Replace “secure” with “safeguard”.

Done

Line 388: After “by 2100” outline what the relative warming would be, as above.

Done. Now Lines 451-452: “radiative forcing reaching 8.5 W m^{-2} (and 4.4°C of warming) ...”

Line 414: What are some examples of these extractive activities relative to each of the levels?

Done

Now Lines 478-479: We then categorized each MPA based on the level of restrictions to extractive activities (recreational, subsistence, and commercial fishing).

And Now Lines 481-485: “This database scores MPAs based on fishing restrictions on a scale of 1–5 scale (1 = Least restricted: no known restriction to the removal of life, 2 = Less restricted: at least one species- or gear-specific restriction apply, 3 = Moderately restricted: several species- or gear-specific restrictions apply, 4 = Heavily restricted: marine life removal is mostly prohibited with a few exceptions, 5 = Most restricted: marine life removal is prohibited).”

Lines 422-423: What are the country-level criteria for fishing restrictions? The authors also mention that they do not include other types of spatial closures and area-based measures that are not MPAs, by assumption this varies from country to country therefore. Some insight should be provided, for example, does this include cultural or seasonal-based closures? MPAs are not all created equal, and this is an important point to be clear on.

Done. We have explained that we followed the same approach as Protected Seas for those missing MPAs from their dataset.

See Now Lines 490-492: “For a few MPAs (34 of 817) that had no LFP score, we reviewed existing information on restrictions on the removal of life and assigned a LFP score following the same approach described earlier.”

For the non-MPAs, we have better framed the text to avoid confusion. We download MPA shapefiles from national agencies’ datasets (see Lines 476 and therefore, the category is based on what nations report as an MPA. For that instance, we did not include other polygons not categorized as MPAs in the country database.

Now Lines 489-490: “We did not include other area-based measures not categorized as an MPA in the national dataset (e.g., fishery management areas).”

Reviewer 3

General Comment

This work focuses on exposure of global floating kelp forests to marine heatwaves and analyze the spatial distribution of kelp forests with respect to the marine protected areas. Although interesting results are presented, major concerns exist, especially in terms of the soundness of methodology and results.

Response General Comment

We thank the reviewer for finding our results interesting and for the concerns they raised, which allowed us to improve the manuscript and clarify parts of the text.

Comment 1

Result completeness. As only the floating (surface canopy) kelp forests can be detected from satellite image, and only 3 of the largest kelp species form floating canopies out of over 120 kelp species, it is not accurate to say that the presented results from satellite observations here are “global kelp forests”, as subsurface kelp forests have not been reasonably accounted for.

Response Comment 1

Done: This point was also mentioned by Reviewer 1, and we apologize for the confusion and for not explaining these points better. We have now made changes to specify that we refer to floating kelp forests, and that the presence of floating kelp habitat, which co-exists with sub-canopy kelps and 100s-1000s of other species (seaweeds, fish, invertebrates) is an indicator of the presence of kelp forest ecosystems, not only a species. Now Lines 105-118 Introduction. We have also expanded the discussion about which areas our maps better represent kelp forest ecosystems and what the gaps are. Now Lines 335-349 . See Response to Reviewer 1 in page 5 and 9.

Comment 2

Classification of floating kelp canopy was derived by applying a globally robust random forest classifier, and the global map was only validated using ground observations at 14 sites in South America. It is largely unknown how the classifier works elsewhere and if the classification accuracy is good enough to support the analysis presented here.

Response Comment 2

Done. We thank the reviewer for their comment, which allowed us to provide more detail on the datasets used, thereby avoiding future confusion. We apologise as we realized the methods poorly explained the kelp maps datasets and missed important details regarding the validation of the global map. The methods sections was restructured many times during internal revisions, and now we realized some of the text could be improved. We have carefully revised this section and improved the text to be more accurate and to remove confusion associated with global classifiers, availability of regional maps, and limitations of the global map. In addition, we now clarify how we gathered the data to integrate the global map.

First, we did not classify floating kelp forests or create new maps, but used existing regionally validated remote-sensed maps to detect floating kelp forests. These maps have been published or uploaded to Kelp Watch following the methods of Bell et al. (2023) (kelpwatch.org) and used similar approaches based on Landsat sensors. For those regions where regional maps were unavailable, we used an existing published global map (Mora-Sota et al. 2020) for floating kelps. We apologize for omitting this detail in the original manuscript, but the authors of this global map first trained their data across eight countries across four continents with existing kelp datasets, then validated their mapping *in situ* across 14 sites from Chile to the Falkland Islands that span 20° of latitude. Finally, they validated their map with previous surveys or observed data at 157 locations in South and North America, New Zealand, Southern Australia and Tasmania, South Georgia, and sub-Antarctic islands (Mora-Soto et al. 2020).

Now Lines in 414-419 Methods: This dataset was validated across 14 in situ sites in South America that cover a variety of coastlines and ecoregions, and with existing data at 151 locations that cover four continents⁵⁰. To ameliorate some potential detection caveats, we excluded pixels that fell within a 30-m buffer relative to the coastline because the global map does not distinguish between intertidal green algae and floating kelp forests and estuaries can also be a source of false positives. See Supplementary Table 1 for the coastlines used to apply the 30-m buffer. “

Comment 3

Results consistency. As Sentinel-2 satellite imagery (2015-2019) was adopted to supplement Landsat observations (1984-98 present), how to guarantee the consistency (or comparability) between results from different satellite data, especially in terms of different band configuration and algorithms used to kelp forest?

Response Comment 3

Both datasets (Sentinel-2 and Landsat) have similar spectral coverage, and since kelp canopy has such a different spectral signature than water (especially in the near infrared), the classification algorithms should produce comparable results for a presence-absence classification. Most importantly, both datasets have been comprehensively validated (see response to above comment). The difference in temporal coverage of the datasets may lead to regional differences and we have acknowledged and clarified this in the text. Specifically, the Sentinel-2 dataset produces a composite over a shorter time period and so may represent an underestimate of potential kelp habitat as compared to Landsat. However, this would only affect our absolute estimates of kelp areas, it should not affect the percent of kelp forests inside/outside MPAs within a region, as the same dataset was used for the entire region and the percentage is relative to the coverage.

Now Lines 350-353 in the Discussion: “We also note that our compiled map may underestimate floating kelp **habitat** for those regions where **regional** maps are not yet available (e.g., Canada, Chile, New Zealand) **because the global map covers a shorter time period than the regional Landsat data and so may miss kelp habitat that was not present between 2015-2019⁵⁰**”

Comment 4

Conclusion soundness. A key conclusion in Abstract is that “...exposure to projected marine heatwaves will increase ~8 times compared to contemporary (2001-2020) exposure for intermediate climate scenarios”. But I didn’t find the corresponding statement in the main text, especially in terms of “the 8 times”. How was it defined and calculated?

Response Comment 4

Done. We have better explained how we obtained those values in the Results. And updated in the abstract, to include the range of scenarios.

Now Lines 173-177 in Results: “**In the long term, even under SSP1-2.6 and SSP2-4.5, these magnitudes are ~5.6 and ~9.6 times higher than contemporary exposure, respectively. Note that these estimates were derived from the mean cumulative MHW intensities (n = 2156 pixels) for each climate scenario and time frame, and then divided by the corresponding mean contemporary values.**”

And Lines 55-56 in Abstract: **projected marine heatwaves will increase >6 to >16 times in the long term (2081-2100) compared to contemporary (2001-2020) exposure.**

Comment 5

Title is not correct and logical. The fact is that intensifying marine heatwaves and anthropogenic activities may threaten global kelp forests, but kelp forests cannot be threatened by protection behavior even if the protection is limited.

Response Comment 5

Done. We have changed the title to reflect the reviewers thought.

New Title: “Global floating kelp forests have limited representation in marine protected areas despite rising threats from marine heatwaves”

Comment 5

In captions of Figure 2 it read SSP-1.26, SSP-2.45, SSP-5.85, whereas in the figure it read different as SSP1-2.6, SSP2-4.5, SSP5-8.5. This issue also exists in the methods parts.

Response Comment 5

Thank you so much, Reviewer 3 also pointed out this error and we have addressed.

Comments Reviewer 4

General Comment

This ms uses remotely sensed data to quantify the distribution of canopy-forming kelp across the world's oceans before overlaying that with MPA designations under different levels of fisheries restrictions. They then calculate cumulative MHW statistics for contemporary times and under different climate forcing scenarios for near and far timescales. They show that kelp regional differences in the susceptibility to MHWs into the future, but also that because the vast majority of kelp forests are not highly protected there will be reduced resilience to MHWs. Given the important role that kelp forests play in supporting elevated levels of diversity as well as providing extensive ecosystem services the impacts of increased MHWs and the lack of protection will have wide ranging consequences. I think this is a timely paper which brings together extensive datasets to undertake the analyses presented. I think that in places the text needs to be toned down and caveated (see specific comments below), but the conclusions are generally sound and the call for better protection for kelp forests is welcomed.

Response General Comment

Thank you so much for finding our work timely, complete and that the conclusions are sound and relevant for kelp forest conservation

Specific comments

Line 50 I don't think there is evidence that climate change poses the greatest risk and your analysis suggests some area may be more at risk from others. I would rephrase.

Done. We have slightly modified the abstract and introduction to ensure its clear. The statement is based on the IPCC Sixth Assessment Report that identified kelp forests as the second most at risk ecosystem from MHWs, after corals.

Now Lines 52 in Abstract: "ecosystems and are **at great risk from climate change"**

Now Lines 79-80 in Introduction: "The IPCC Sixth Assessment Report identified** kelp forests **as the second most at-risk** marine ecosystems from MHWs⁶... "**

Line 51 I would rephrase the last part of this sentence as there is quite a lot of literature on the future threats to kelp and at least in a regional context and understanding of the current conservation status (or lack thereof). Suggest rephrasing

Done. We have added "global" before "future threats and conservation status" to ensure its clear that we refer to their threats and conservation globally, not regionally.

Abstract – Suggest adding floating to all discussion of kelp in the abstract as it suggests as written that the outputs are extrapolated to all kelps, which isn't the case and also not possible with this analysis.

Done

Line 58 "the most effective conservation measure for providing climate resilience" I find this

phrase a little difficult throughout. Yes highly restrictive marine protected areas that stop habitat loss and ensure intact predators in kelp systems controlled by grazers are likely to increase climate resilience, but if the chief driver of kelp loss is say invasive species/ eutrophication (as it is in some parts of the world) this type of MPA would add little to make kelp forests for resilient to climate change. I would suggest using more nuanced language.

Done. Thank you. This point was also raised by reviewer 1. Given the limited space in the abstract to explain when MPAs may provide resilience, we have changed this sentence to read more general regarding the efficiency of fully protected MPAs for biodiversity protection. See Response to Reviewer 1, page 2.

Lines 71-72 I suggest rephrasing this passage as we generally have a good global understanding of kelp forests distributions and an understanding of responses to climate change and MHWs from in-situ studies and experiments. I agree at regional scales kelp forest extent can be much more poorly resolved.

Done. Thank you for your comment, which allowed us to clarify our point. Here, we mean “spatial data”. That means accurate and comprehensive maps of kelp distribution, from which we can then conduct spatial analyses of threats, protection, and so on.

Now Lines 43: “Comprehensive maps on kelp forest distribution....”

Line 74 Productivity is more similar to broadleaf forests than rainforests or coral reefs. See Pessarrodona et al (2022) Science Advances eabn2465 – Fig 1a.

Done, we have changed the sentence to say” comparable to terrestrial forests” and cited pessarrodona et al. 2022. Now Line 77.

Line 76-77 I am not convinced by this line “Kelp forests are among the marine ecosystems at greatest risk from MHWs”. Surely this depends on where the MHW occurs in the species range and I would suggest that there is equal evidence that other species and ecosystems (e.g. coral reefs) are equally or more at risk. I suggest toning the language.

Done. See previous comment in page 14. Here we cite the six IPCC report.

See Now Lines xx

Line 78 I am not suggesting that MHWs haven't contributed to the loss of kelp in Tasmania, but the key driver is expansion of urchins (as a result of decadal scale warming) and over fishing of large rock lobste. Suggest altering the language or removing Tasmania as an example of MHW driven kelp loss.

Done. We have modified the text and excluded Tasmania from the example, as suggested by the reviewer. Now Lines 83-84: “For example, northern California **has lost >90% of its kelp forests due to the combined effects of severe marine heatwaves and overgrazing by sea urchins^{23,24}.”**

Line 106 I know that you have defined above that by kelp forests you mean canopy-forming

forests, but in places throughout the ms I think it would be useful to emphasise that the ms is only referring to canopy-forming forests and I think this is one of those instances.

Done

Line 136 Should be Fib 1b and 2 referred to here

Done

Line 145 Suggest adding temperate South Africa here as they look very similar to temperate South America.

Done

Figure 2b I suggest using the same labelling system for the SSP as in the legend i.e. the – after SSP. Also I think that the SSPs are incorrectly labelled. At present SSP – 1.26 has the most extreme cumulative MHWs. Finally I suggest changing the colouring for contemporary and (currently labelled) SSP – 5.85.

Done. Thank you for catching the mislabeling for the SSP in the legend text, we have now changed it to be the same as in the figure (e.g., SSP1-2.6). In addition, we have updated Figure 2b and changed the mislabeled SSP1-2.6 and SSP5-8.5. We use IPCC standard colors for labelling SSPs, therefore we prefer to stick with these colors.

Line 173 Suggest emphasising floating kelp forests here

Done

Line 186 It might be worth some discussion in the ms about whether these are real or paper HPAs, particularly in the Southern Ocean, given their isolation (maybe some discussion of this in the section beginning line 304). Also what does this 99% protection represent in terms of global kelp? If 99% of small amount of kelp forest this doesn't really mean anything. Might be worth including this information for context.

Done. The discussion about the Southern Ocean realm (not the Southern Ocean) being isolated and likely less threatened by human activities (Paper parks are places not well managed or regulated, not isolated or less threatened) is touched on and referenced in the first paragraph of the Discussion (Lines 276-278). However, we add a bit more detail regarding the effectiveness of MPAs.

Now Lines 364-367 in the discussion: Therefore, including such information will likely decrease the coverage of kelp forests within MPAs with high levels of effective protection because many lack effective governance, enforcement, or community involvement⁷⁰.”

We have added this text to clarify how much that 99% represents globally.

Now Lines 223-224 in Results: “has 99.9% of its kelp forests within highly protected MPAs (which represents 11% of the global distribution, see Fig. 3c, Fig. 4).”

Also see: Figure 3c, (protection decreases from 13.7% to 2.8% when excluding these areas), described in Results (Lines 209-212) and discussion (Lines 276-282). In addition,

Figure 4 shows contribution of all realms and countries to the global coverage of floating kelp.

Figure 3 may be worth also including the island/islands that are protectorates of France rather than just stating France. Maybe consider the same for UK protectorates although these do come out in the text.

Done. We have added text.

Legend Fig 3, Lines 236: France (Kerguelen and Crozet Islands)

Legend Fig 4, Lines 247-249: “Note that mainland France and the United Kingdom have no floating kelp forests in their mainland and are all in their overseas territories in the Southern Ocean realm.”

Using level of fishing protection is only a good measure of protection if grazing is a key pressure and more intact fish stocks reduce this or that destructive fishing gears are removed which reduces habitat loss. Where grazing by urchins may not be a key driver these HPMPAs may not increase resilience. Is there confidence that grazing is a key driver across all these regions? It isn't in all kelp forests and if there isn't confidence then I think that this needs caveating in the ms – maybe in the methods, but potentially also in the main ms (maybe around lines 245-246?).

Done. Great point, that has been brought up by Reviewer 1 and has helped us clarify and improve the manuscript. Please see response to Reviewer 1, page 2, and Lines now 86-101.

The identification of refugia is made in the abstract and line 277, but I am somewhat sceptical that areas of lower MHW impact will be able to act as a refugia given the distance between the most and least impacted areas. The identified refugia are in the Southern Ocean while the most impacted are in the Pacific Arctic. *Macrocystis* has greater dispersal ability than many kelps, but this seems to be stretching the resupply from refugia further than perhaps it should go. I believe this needs some discussion and toning down.

Done. We are not suggesting that these areas will be climate refugia to support recovery of global floating kelp populations which are 1000-10,000 kms apart. Our analyses identify areas less exposed to future MHWs, and thus may provide climate refugia for those regions. Following comments by Reviewer 1, we have modified parts of the abstract and introduction to clarify.

Now Lines 56-59 in Abstract: “While exposure will intensify across all regions, some southern hemisphere areas which have lower exposure to contemporary and projected marine heatwaves **may provide climate refugia for floating kelp forests.”**

And Lines 122-124 in Introduction: “may enhance the resilience of other kelp forests, **depending on local and regional connectivity and live history-traits, by maintaining a source of recovery for impacted kelp habitats²⁴.”**

Lines 292-295 It is stated that there will be some overlap with floating and subsurface kelp and therefore the outcomes may be similar for kelp in these regions. I agree, but I think it is worth stating that whole ocean basins e.g. NE Pacific and N Atlantic with kelp present are not represented at all as they don't possess floating kelp.

Done. We agree with the reviewer and as suggested by Reviewer 1, we have made this point clear. See response to reviewer 1 comments in page 5 and 9. Now in Lines 105-118 in the Introduction, and Lines 335-349 in the Discussion.

Line 366 It isn't clear to me what the filters and masks are in Supplementary Table 1

Done. Now Line 419: “for the coastlines used to apply the 30-m buffer.”

Line 368-369 Given the evidence for regional losses of kelp over the last 20-30 years I am a little sceptical of using Landsat images back to 1984. What if all the detections were early in the dataset and there have none in the later years? Was the potential for range reductions in kelp forests over the time-series considered? If it wasn't I suggest that some sort of temporal analysis of this should take place before using data back to 1984.

Done. We have better clarified why we used all available data in the Methods.

Now lines 421-429: Our final floating kelp habitat map includes any pixel for which the satellite detected kelp in the time series and represents the known presence of floating kelp in the timeseries. We included all observations to avoid arbitrarily choosing a period to map the distribution of floating kelp forests because these ecosystems are highly dynamic⁷⁴ and we were interested in detecting potential kelp habitat. For example, in some places kelps may be expanding their distribution (cold-edges)^{56,57}, while in others places kelps may be in alternative stable states dominated by urchin barrens²³ (degraded kelp ecosystems) or by more heat tolerant subcanopy kelp species (competing with floating kelps) near warm-edges⁷⁸. These alternative stable states can shift, even after decades^{79,80}.

Line 373 I suggest that you make the length and years of the baseline period clear here. It is mentioned below, but it would be best to mention when first stated.

Done. Now Lines 432-433 in Methods: “projected cumulative annual MHW intensities for each kelp pixel using a baseline climatology of 1983–2012.”

Line 398 It isn't clear to me why a different base line would be used for the reference period and the climatology. When not use the same period?

We have revised the wording in the Methods to make this clearer. We use only OISST (observational) temperatures to assess contemporary marine heatwaves. We use only CMIP6 ESM data to assess future marine heatwaves. The latter is necessary because ESMs do not necessarily faithfully represent daily variability, so projected MHWs should be assessed on a like-for-like basis against data from ESM historical runs.

Now Lines 442-444: “For each ESM, we selected the Historical run to represent the recent past (1983–2014), and selected three future (2015–2100)...”

And Now Lines 452-455: “We bias-corrected the SST dataset from each ESM relative to the corresponding ensemble mean of the Historical runs using the delta method (see⁸³). This method ensures that projections for each ESM blend smoothly to the end of the Historical runs for the reference period 1983–2014.”

Lines 416-419 I think it would be helpful what the different categories mean in terms of restrictions and LFP.

Done. Reviewer 1 also suggested and we have now address. See Page 10, and Now Line 478-485 .